



# A global zircon U–Th–Pb geochronology database

Yujing Wu [1], Xianjun Fang [1], Jianqing Ji [1, *]

[1]School of Earth and Space Sciences, Peking University, Beijing 100871, China

*Correspondence to*: Jianqing Ji (grsange@pku.edu.cn)

**Abstract.** Since the start of the 21st century, the widespread application of ion probes has promoted the mass output of high-precision and high-accuracy U–Th–Pb geochronology data. Zircon, as a commonly used mineral for U–Th–Pb dating, widely exists in the continental crust and records a variety of geological activities. Due to the universality and stability of zircons and the long half-lives of U and Th isotopes, zircon U–Th–Pb geochronology can provide nearly continuous records for almost the entire history of Earth and is thus essential to studying the growth and evolution of the continental crust and even Earth system
evolution. Here, we present a database of zircon U–Th–Pb geochronology that samples the global continental crust and spans nearly all of Earth's history. This database collects ~2,000,000 geochronology records from ~12,000 papers and theses and is by far the largest geochronology database to our knowledge. This paper describes the complied raw data, presents the relationship between dating error and zircon age, compares the error levels of different dating methods, and discusses the impact of sampling bias on data analysis as well as how to evaluate and weaken this impact. In addition, we provide an
overview of the temporal and spatial distribution of global zircon ages and provide key insights into the potential research value of zircon ages for Earth system science, such as crustal evolution, supercontinent cycles, plate tectonics, paleoclimate changes, biological extinction, as well as commercial use in mining and energy. Overall, this collection not only provides us with a comprehensive platform with which to study zircon chronological data in deep time and space but also makes it possible to explore the underlying geodynamic mechanisms and evolution of Earth's system and its astronomical environment.

## 1. Introduction

Zircon U–Th–Pb geochronology has been made much more practical since Krogh (1973) invented isotope dilution-thermal ionization mass spectrometry (ID-TIMS) (Song, 2015; Davis et al., 2003). With the widespread use of ion probe mass spectrometers, in situ microanalysis can be performed efficiently and precisely, promoting the rapid development of zircon U–Th–Pb dating (Gehrels, 2014; Carrapa, 2010). The main dating methods at present include (Becker, 2007) laser ablation
inductively coupled plasma–mass spectrometry (LA-ICP–MS), secondary ion mass spectrometry (SIMS), sensitive high resolution ion microprobe (SHRIMP), and thermal ionization mass spectrometry (TIMS).

The U–Th–Pb decay system plays a key role in geochronology. The half-lives of $^{238}$U, $^{235}$U, and $^{232}$Th isotopes are long enough to date Earth's entire history but short enough to allow for the accurate measurement of both parent and daughter isotopes. Based on the decay of $^{238}$U→$^{206}$Pb (half-life: 4.47 billion years (Gyr)), $^{235}$U→$^{207}$Pb (0.70-Gyr half-life), and
$^{232}$Th→$^{208}$Pb (14.01-Gyr half-life) (Jaffey et al., 1971), we can obtain three ages, i.e., $^{206}$Pb/$^{238}$U, $^{207}$Pb/$^{235}$U, and $^{208}$Pb/$^{232}$Th ages. Using the abundance ratio of natural U isotopes, $^{238}$U/$^{235}$U≈137.8, another $^{207}$Pb/$^{206}$Pb age can be derived (Spencer et al., 2016; Hiess et al., 2012). If these ages are consistent with one another, the decay system is closed, verifying the reliability of the measured ages. This also represents the advantage of U–Th–Pb dating over other isotope dating methods.

Zircon is a common accessory mineral that can stably exist in various kinds of rocks and is distributed throughout the
global continental crust of all ages (Hanchar and Hoskin, 2018; Hawkesworth et al., 2010). It is not unusual for zircon to survive through multiphase complex magmatism and metamorphism as a result of its physical and chemical stability (Hawkesworth et al., 2010). Due to its low original Pb content, rich Th and U contents, and high closure temperature for trace element diffusion, zircon is one of the most widely used minerals for U–Th–Pb isotopic dating (Williams, 2015). In addition, the zircon age distribution can span nearly all of geological history. The oldest zircon discovered thus far is 4.4 Gyr old (Wilde
et al., 2001). Thus, zircon U–Th–Pb geochronology provides an excellent means to explore the deep-time evolution of the continental crust (Voice et al., 2011).



With the widespread application of ion probe mass spectrometers, a large number of zircon U–Th–Pb ages have been measured by various chronology laboratories around the world in the past two decades (Puetz and Condie, 2019; Wu et al., 2022b). These zircons are sampled in the global continental crust, with ages nearly continuously spanning from 4.4 billion

years ago (Ga) to the present (Puetz et al., 2017). However, in most cases, these zircon samples were used for independent regional studies and would probably not be used thereafter (Wu et al., 2019). We believe that if we could collect zircon U–Th–Pb dating records for the past decades to build a database, we would be able to derive the zircon production history of the global continental crust and then study the related magmatism, metamorphism, and sedimentary processes. Furthermore, we could explore the evolution and distribution of Earth's interior and surface processes on time and space dimensions. A global

zircon database would not only be of great value to academic research but would also have potential commercial uses in mining and energy (oil and gas).

Many scholars have previously collected zircon data to explore the evolution of solid Earth and geodynamic history. Voice et al. (2011) compiled ~5100 individually dated detrital zircon samples with ~200,000 dating records and suggested that the temporal distribution of zircon ages could indicate episodic crustal recycling. This evidence provided by zircons is

consistent with plate tectonics. By analyzing the time distribution of zircon U–Th–Pb ages, combined with craton collisions and crustal cycles, Condie (2013) explored the evolution of the Proterozoic crust from the Nuna supercontinent to the Rodinia supercontinent. Mckenzie et al. (2016) used ~120,000 detrital zircon U–Pb age data from areas around the world to explore the spatial distribution of continental magmatic arc systems in the Cryogenian period. Puetz and Condie (2019) collected U–Pb age data for ~610,000 detrital zircons and ~212,000 igneous zircons sampled from around the world, as well as 5 other

isotope databases, showing the geochemical cycles of mantle evolution. Wu et al. (2022b) analyzed the zircon production series of the global continental crust, compared it to astronomical driving factors, and proposed that the evolution of the continental crust and even the earth system may be affected by the astronomical environment in which the earth is located.

However, if the amount of data is not sufficient, the resolution of zircon age series will be lower, leading to possible analysis bias. In addition, limited sampling locations will also affect the objectivity of statistics (Wu et al., 2020; Puetz et al.,

2017). Here, we present a database of ~2,000,000 zircon U–Th–Pb dating data sampled from the global continental crust (Wu et al., 2022a). Dating methods used include LA-ICP–MS, SHRIMP, SIMS, and TIMS; zircon host rocks include igneous, metamorphic, and sedimentary rocks. Undoubtedly, this database provides a more comprehensive and objective chronology data source on both the time and space dimensions for future earth system science research. From this database, scholars can not only obtain an overview of global zircon production throughout Earth's entire history but also study the evolution of zircon

production in each period and region. If combined with other geological events and astronomical environments, it is also possible to deeply explore the energy source of Earth's dynamics and the mechanisms behind it. In the future, this database may even provide constraints for astronomical parameters and their evolution, expanding the deep-time dimension of astronomical parameters. In addition, from the perspective of data science, the large data volume and global sampling range of this database give us a good experimental platform for analyzing and solving biased sampling issues (hot data issues).

**2. Data**

Here, we collected ~2,000,000 zircon U–Th–Pb age data sampled from the global continental crust from ~12,000 references (Wu et al., 2022a), with zircon ages spanning all of Earth's history. This database is based on the original Chinese zircon U–Th–Pb dating database (Fang et al., 2018; Wu et al., 2019), from which the data were updated for nearly three years and sampling sites were expanded to the global continental crust. The compilation of zircon data is divided into two categories: 1)

the "Database" files ("Database_part1.xlsx" and "Database_part2.xlsx"), which include ~2,000,000 records of zircon U–Th–Pb ages and their sample information, dating methods, parent rock lithology, sampling locations, and reference numbers, and 2) the "References" file ("References.xlsx"), which provides information on ~12,000 original references corresponding to the zircon age data, including author, year of publication, and publication detail data. For each record in the "Database" files, its reference can be found in the "References" file from its reference number. This database is available at

https://doi.org/10.5281/zenodo.7387567.

It should be noted that we improved the codes for extracting GPS information, so the GPS data used here are slightly



different from the sampling map given in Wu et al. (2022b) (see Figure 1). The improvement details are as follows: 1) In the original references, the symbols of GPS "degrees, minutes and seconds" are various, and it is difficult for a computer to completely distinguish them, and sometimes they are not even recognized or are recognized incorrectly; the new modification

expands the recognition range of the codes for "degrees, minutes and seconds" symbols and manually normalizes some abnormally identified GPS data according to the original references. 2) The initial GPS extraction codes were aimed at the Chinese continental crust, and the GPS range was restricted in the screening step. Later, when the code was updated to extract the global GPS, some details were omitted, resulting in the loss of some GPSs, especially for Europe.

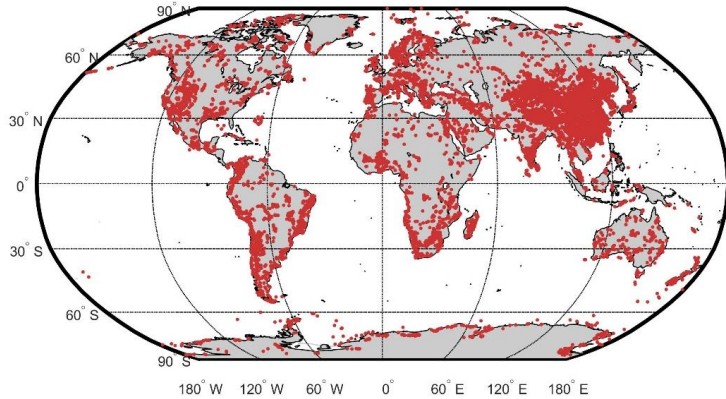

**Figure 1. Map of the distribution of zircon sampling locations. Each red dot indicates a sampling location (adapted from Wu et al. (2022b)).**

### 2.1 Geochronology database

In the "Database" files, fields of each record are defined as follows:

  (01) *Ref_number*: the reference number corresponding to multiple chronology records.

100   (02) *Data_number*: the number of each chronology record, which is a unique index.

  (03) *Sample_number*: the corresponding rock sample or zircon sample number for each chronology record.

  (04) *Method*: the dating method, including LA-ICP–MS, SHRIMP, SIMS, TIMS, etc.

  (05) *age206Pb/238U*: the mean age derived by $^{238}$U-$^{206}$Pb decay.

  (06) *age206Pb/238U_σ*: the 1σ absolute error (standard deviation) of the $^{206}$Pb/$^{238}$U age.

105   (07) *age207Pb/235U*: the mean age derived by $^{235}$U-$^{207}$Pb decay.

  (08) *age207Pb/235U_σ*: the 1σ absolute error (standard deviation) of the $^{207}$Pb/$^{235}$U age.

  (09) *age207Pb/206Pb*: the mean age derived based on the decay of $^{238}$U-$^{206}$Pb and $^{235}$U-$^{207}$Pb, and the abundance ratio of $^{238}$U/$^{235}$U.

  (10) *age207Pb/206Pb_σ*: the 1σ absolute error (standard deviation) of the $^{207}$Pb/$^{206}$Pb age.

110   (11) *age208Pb/232Th*: the mean age derived by $^{232}$Th-$^{208}$Pb decay.

  (12) *age208Pb/232Th_σ*: the 1σ absolute error (standard deviation) of the $^{208}$Pb/$^{232}$Th age.

  (13) *Lithology*: the lithology of zircon's host rock, including sedimentary, igneous, and metamorphic rocks.

  (14) *Longitude*: the longitude of the sampling site.

  (15) *Latitude*: the latitude of the sampling site.

115   In some chronological records in the "Database" files, some of the fields mentioned above are empty because the relevant dating and sample information are not given in the original literature. For each chronological record in the "Database," the source references can be found in the "References" file by *Ref_number*.



**Table 1. Data specifications of the zircon U–Th–Pb database.**

| Field name | Field description | Data details |
|---|---|---|
| *Ref_number* | Reference number | The number of the reference corresponding to each chronology record. |
| *Data_number* | Record number | The number of each chronology record. |
| *Sample_number* | Sample number | The number of each rock sample or zircon sample corresponding to each chronology record. |
| *Method* | Dating method | Including LA-ICP–MS, SHRIMP, SIMS, TIMS, etc. |
| *age206Pb/238U* | Age derived from $^{206}Pb/^{238}U$ | Unit: Myr |
| *age206Pb/238U_σ* | Standard deviation of $^{206}Pb/^{238}U$ age | Unit: Myr |
| *age207Pb/235U* | Age derived from $^{207}Pb/^{235}U$ | Unit: Myr |
| *age207Pb/235U_σ* | Standard deviation of $^{207}Pb/^{235}U$ age | Unit: Myr |
| *age207Pb/206Pb* | Age derived from $^{207}Pb/^{206}Pb$ | Unit: Myr |
| *age207Pb/206Pb_σ* | Standard deviation of $^{207}Pb/^{206}Pb$ age | Unit: Myr |
| *age208Pb/232Th* | Age derived from $^{208}Pb/^{232}Th$ | Unit: Myr |
| *age208Pb/232Th_σ* | Standard deviation of $^{208}Pb/^{232}Th$ age | Unit: Myr |
| *Lithology* | The lithology of the host rock | Including igneous, sedimentary, and metamorphic rocks. |
| *Longitude* | The longitude of the sampling location | Unit: degree; range: -180 to 180. |
| *Latitude* | The latitude of the sampling location | Unit: degree; range -90 to 90. |

**Note: "Myr" indicates "million years."**

## 2.2 Data references

The chronological data used in this study were collected from ~12,000 references of the following academic publishers: Elsevier (ScienceDirect full-text database), Cambridge, Geological Society of London, Oxford, Springer, Taylor & Francis, Wiley, and China National Knowledge Infrastructure (CNKI). Original reference information for the chronological data is included in the "References" file (see the data repository on Zenodo). The fields of each reference record are detailed in Table 2. The field "*Ref_number,*" namely, the reference number, cites the unique index in the "References" file, corresponding to multiple chronological records in the "Database" files.

**Table 2. Data specifications of the References file.**

| Field name | Field description |
|---|---|
| *Ref_number* | Reference number |
| *Author_surname* | The surname of the first author |
| *Author_given_name* | The given name of the first author |
| *Year_publication* | Year of paper publication |
| *Journal* | The journal in which the paper was published |
| *Volume* | The volume of the paper |
| *Issue* | The issue of the paper |

**Note: For references in Chinese, the "*Author_surname*" field is the full name of the first author owing to the Chinese citation format.**

## 3. Data characteristics and distribution

### 3.1 Dating error vs. age

The zircon U–Th–Pb dating error is related to the zircon age. By analyzing a large number of zircon ages and their errors, we can obtain the dating error curves of the four age types with age and select the age type with the smallest error as the recommended age.

In this study, the age error within a certain time interval is obtained using the moving average method (Wu et al., 2022b).
The length of the sliding window (bin size) is the median of all age errors (14 Myr), and the sliding step size is 1 Myr. For each age type, when the window slides along the time axis, the median of the age error in each window is counted and assigned to the middle age of the window. Then, we can obtain an error-age scatter diagram (see Figure 2). It is obvious that for the three ages of $^{206}$Pb/$^{238}$U, $^{207}$Pb/$^{235}$U, and $^{208}$Pb/$^{232}$Th, the age error is positively correlated with the age, and at ages of above ~3200 Ma, the error level begins to fluctuate and increase greatly; for the $^{207}$Pb/$^{206}$Pb age, the age error is negatively correlated
with the age as a whole, and the error level begins to fluctuate and increase greatly at ages of above ~3800 Ma. Given the error trends of the 4 age types, the recommended ages for young, middle, and old age samples are $^{206}$Pb/$^{238}$U, $^{207}$Pb/$^{235}$U, and $^{207}$Pb/$^{206}$Pb ages, respectively. When the sample is too old, the dating uncertainty of all age types increases significantly, which may be attributed to sample preservation.

Then, the error points for various ages were fitted to obtain error curves (see the equations in Table 3). To exclude the
impact of the great increase in dating uncertainty due to old age on data fitting, the fitting of $^{206}$Pb/$^{238}$U, $^{207}$Pb/$^{235}$U, and $^{208}$Pb/$^{232}$Th ages is conducted for 0-3200 million years ago (Ma) and that of $^{207}$Pb/$^{206}$Pb age is conducted for 0-3800 Ma (Figure 2 and Table 3). According to the error fitting curves given in Table 3, we can calculate the intersection points of the curves and give the recommended age for each age interval: 0-1163 Ma, $^{206}$Pb/$^{238}$U age; 1163-2390 Ma, $^{207}$Pb/$^{235}$U age; and >2390 Ma, $^{207}$Pb/$^{206}$Pb age (Table 4).

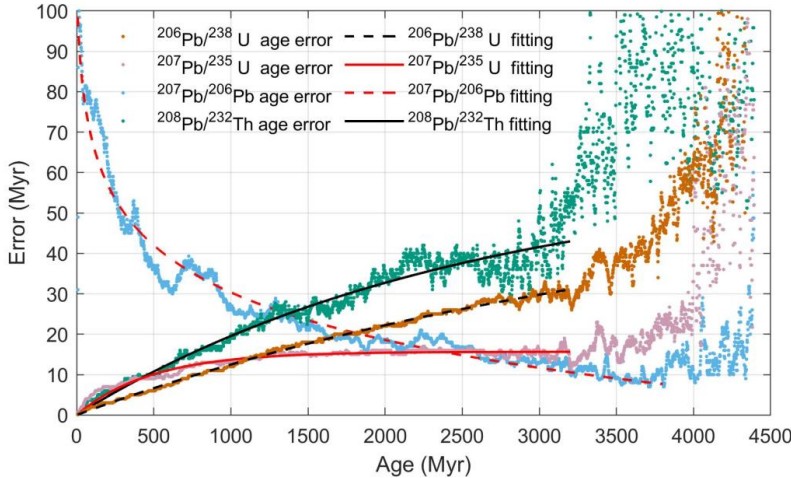


**Figure 2. Age errors of four ages and their fitting curves.**

**Table 3. Error fitting curves for different age types.**

| Age type | Regression equation | Adjust R-Squared | Parameters with 95% confidence interval (CI) |
|---|---|---|---|
| $^{206}$Pb/$^{238}$U | y=a*[1−exp(−b*x)] | 0.9957 | a = 56.32  (55.69, 56.95) |
|  |  |  | b = 2.507  (2.470, 2.545) ×10$^{-4}$ |
| $^{207}$Pb/$^{235}$U | y=a*[1−exp(−b*x)] | 0.9417 | a = 15.66  (15.62, 15.70) |
|  |  |  | b = 2.066  (2.040, 2.091) ×10$^{-3}$ |
| $^{207}$Pb/$^{206}$Pb | y=a−b*ln (x+c) | 0.9478 | a = 148.7  (147.3, 150.1) |
|  |  |  | b = 17.11  (16.92, 17.30) |
|  |  |  | c = 11.90  (8.996, 14.80) |
| $^{208}$Pb/$^{232}$Th | y=a*[1−exp(−b*x)] | 0.9416 | a = 59.90  (58.50, 61.29) |
|  |  |  | b = 3.948  (3.803, 4.093) ×10$^{-4}$ |

**Note: Variable "x" denotes age. Variable "y" denotes age error.**

Data

## 3.2 Comparison of dating methods

Zircon dating errors vary not only with age but also with dating methods. Although TIMS is more precise, other methods are more efficient and widely used (Gehrels, 2014). This paper gives the relationship between the error and age of the four dating methods of LA-ICP–MS, SHRIMP, SIMS, and TIMS. The numbers of chronological records for the methods above are $1.6\times10^6$, $2.6\times10^5$, $8.0\times10^4$, and $3.2\times10^4$, respectively. The relationships between the various age errors of these four dating methods are similar, but the specific intersection points of the curves are different (see Figures 3-7 and Table 4). The intersection points of the error curves of SIMS and TIMS are closer to the empirical rule for selecting the recommended age; that is, $^{206}Pb/^{238}U$ age is the recommended age for < 1000 (800-1200) Ma and $^{207}Pb/^{206}Pb$ age for >1000 (800-1200) Ma. However, since approximately 80% of the chronological records are dated using LA-ICP–MS, the age error curves of the entire database are closer to those of the LA-ICP–MS, which indicates that the previous empirical rules must be used with caution.

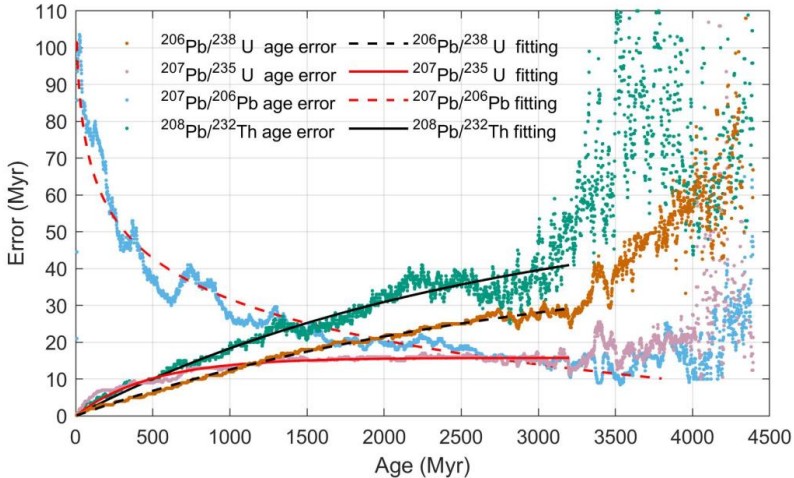

Figure 3. Error fitting curves of different ages derived from LA-ICP–MS.

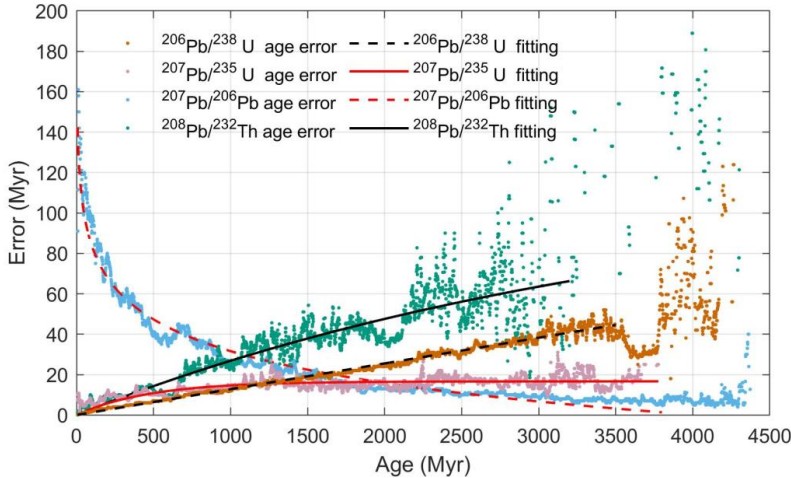

Figure 4. Error fitting curves of different ages derived from SHRIMP.



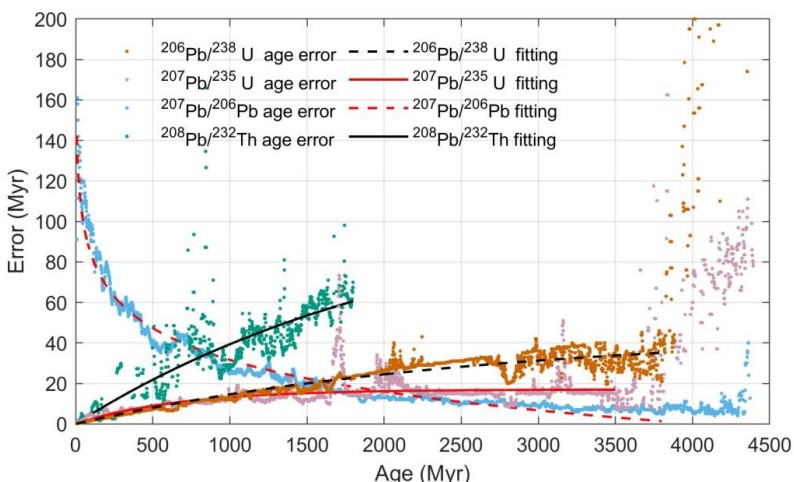

**Figure 5. Error fitting curves of different ages derived from SIMS.**

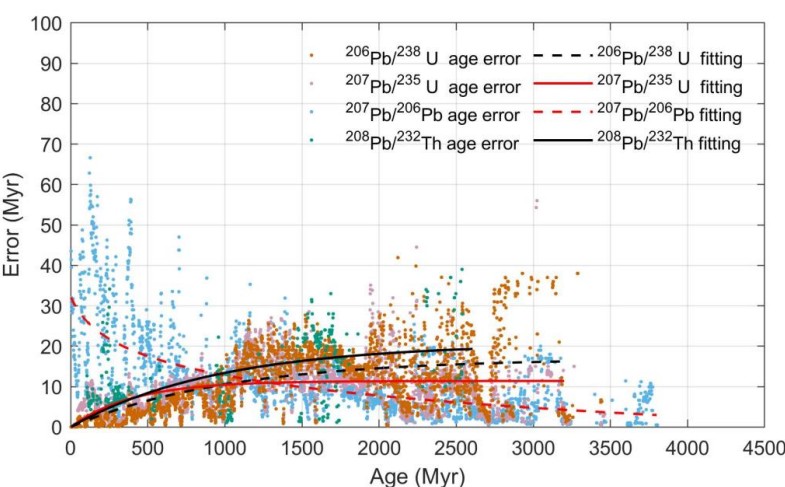

**Figure 6. Error fitting curves of different ages derived from TIMS.**

**Table 4. Intersection of age error curves (unit: Ma)**

| Intersection of age error curves | All methods | LA-ICP–MS | SHRIMP | SIMS | TIMS |
|---|---|---|---|---|---|
| $^{206}Pb/^{238}U$ and $^{207}Pb/^{235}U$ | 1162.54 | 1177.34 | 1197.74 | 748.07 | 971.38 |
| $^{206}Pb/^{238}U$ and $^{207}Pb/^{206}Pb$ | 1795.50 | 1933.79 | 1619.01 | 1273.76 | 1182.52 |
| $^{207}Pb/^{235}U$ and $^{207}Pb/^{206}Pb$ | 2390.44 | 2694.25 | 1956.91 | 1548.17 | 1295.16 |



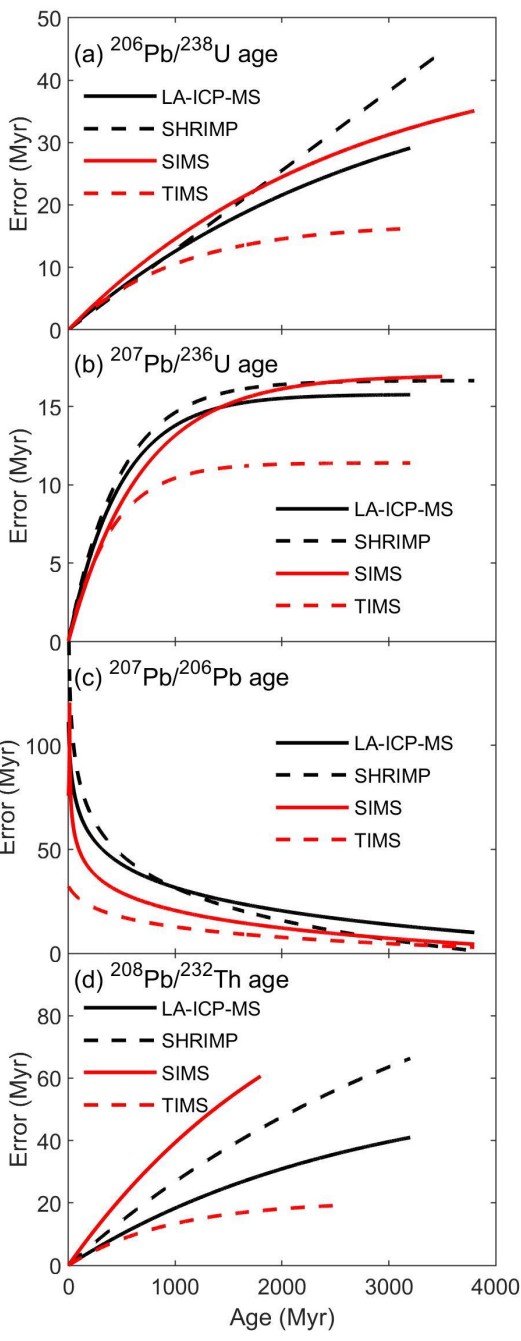

**Figure 7. Comparison of age errors of different dating methods. (a) $^{206}$Pb/$^{238}$U age; (b) $^{207}$Pb/$^{235}$U age; (c) $^{207}$Pb/$^{206}$Pb age; (d) $^{208}$Pb/$^{232}$Th age.**



**Table 5. Age error fitting of different dating methods.**

| | LA-ICP-MS | | SHRIMP | | SIMS | | TIMS | |
|---|---|---|---|---|---|---|---|---|
| Age type | Adjust r² | Parameters with 95% CI | Adjust r² | Parameters with 95% CI | Adjust r² | Parameters with 95% CI | Adjust r² | Parameters with 95% CI |
| $^{206}Pb/^{238}U$ | 0.9935 | a = 44.51 (44.07, 44.94) b = 3.318 (3.270, 3.365) ×10⁻⁴ | 0.9842 | a = 3568 (-2544, 9680) b = 3.591 (-2.590, 9.771) ×10⁻⁶ | 0.896 | a = 45.76 (44.6, 46.93) b = 3.830 (3.666, 3.994) ×10⁻⁴ | 0.3756 | a = 17.00 (16.20, 17.80) b = 9.680 (8.537, 1.082) ×10⁻⁴ |
| $^{207}Pb/^{235}U$ | 0.9370 | a = 15.77 (15.72, 15.81) b = 2.071 (2.044, 2.097) ×10⁻³ | 0.4956 | a = 16.65 (16.53, 16.78) b = 2.102 (2.023, 2.181) ×10⁻³ | 0.3863 | a = 17.00 (16.70, 17.30) b = 1.484 (1.384, 1.585) ×10⁻³ | 0.1674 | a = 11.40 (11.11, 11.69) b = 2.465 (2.168, 2.762) ×10⁻³ |
| $^{207}Pb/^{206}Pb$ | 0.9250 | a = 143.9 (142.4, 145.5) b = 16.23 (16.02, 16.43) c = 6.459 (3.653, 9.266) | 0.9517 | a = 188.3 (186.8, 189.9) b = 22.69 (22.48, 22.90) c = 0.5763 (-0.9437, 2.096) | 0.9136 | a = 103.9 (102.8, 105.0) b = 12.07 (11.92, 12.22) c = -11.25 (-13.08, -9.422) | 0.4027 | a = 66.65 (61.49, 71.81) b = 7.705 (7.043, 8.368) c = 86.60 (38.92, 134.3) |
| $^{208}Pb/^{232}Th$ | 0.9279 | a = 58.61 (56.99, 60.22) b = 3.751 (3.591, 3.911) ×10⁻⁴ | 0.7579 | a = 117.8 (106.2, 129.3) b = 2.585 (2.245, 2.925) ×10⁻⁴ | 0.6576 | a = 111.9 (92.07, 131.7) b = 4.332 (3.303, 5.362) ×10⁻⁴ | 0.1662 | a = 20.68 (17.60, 23.76) b = 10.43 (6.889, 13.97) ×10⁻⁴ |

Note: Regression equations are the same as those in Table 3.

### 3.3 Temporal characteristics of zircon production

Zircon production increases with magmatic and metamorphic activities. Therefore, the amount of zircon production can be used to understand the past intensity of geological activity (Hawkesworth et al., 2010). A simple and direct proxy is the number

of zircon age records for different geological times, which can indicate the intensity of magmatic and metamorphic activities (Wu et al., 2022b). Using the moving average method, we can obtain the time series of zircon production (Figure 8). Figure 8 shows that the zircon age series of all lithologies is more similar to that of sedimentary rocks, which may be explained by zircons in sedimentary rocks being composed of a natural mixture of igneous and metamorphic zircons. Nevertheless, the zircon production peak periods reflected in the age series of sedimentary, igneous, and metamorphic rocks are basically

consistent, i.e., ca. 800, 1000, 1850, 2500, 2700, 3200, and 3400 Ma in the Precambrian. Figure 9 is presented to more clearly show zircon production in the Phanerozoic, during which prominent zircon age peaks occurred ca. 50, 130, 250, 300, and 440 Ma, reflected in the zircons of all three lithologies. However, sedimentary rocks at 100 and 160 Ma; igneous rocks at 220 Ma; and metamorphic rocks at 220, 240, and 500 Ma have different zircon peaks. Specific geological meanings can be analyzed in depth with the help of this database as well as other geological evidence.

Furthermore, Figure 8 and Figure 9 clearly show that the time series of zircon production is multiscale periodic. Wu et al. (2022b) systematically analyzed the periodicity of the zircon age series and finally gave the following cycles: ca. 800, 360, 220, 160, 69, 57, 44, 30, 20, and 17 Myr.

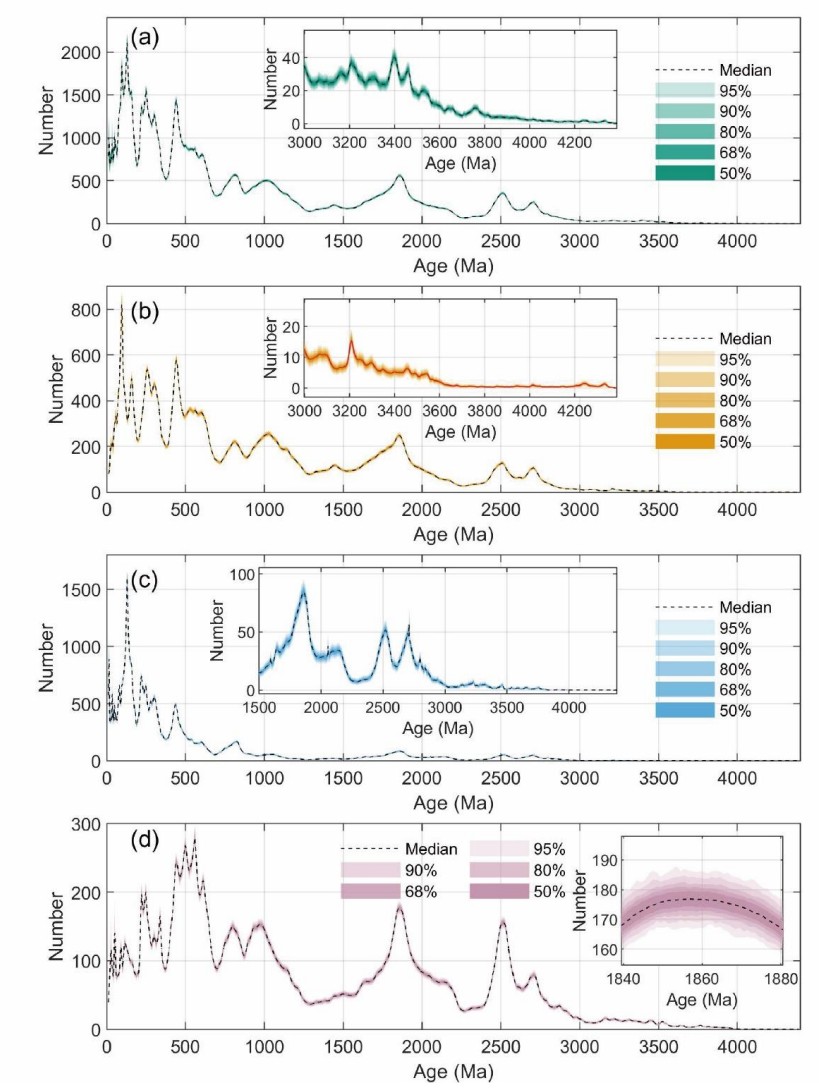

**Figure 8. Zircon production series since 4.4 Ga. The host rocks of each zircon series are (a) of all lithologies, (b) sedimentary rocks, (c) igneous rocks, and (d) metamorphic rocks. The insets of panels (a-c) focus on the data for before 3 Ga. The inset of Panel (d) highlights the impact of dating errors on age series, which can be disregarded, as shown. The filled zones represent 1000 Monte Carlo simulations of the zircon series with different transparencies, indicating different distribution probabilities, as shown in the legend. The dashed line indicates the median of the simulations. In each simulation, simulated zircon ages are selected based on their dating error. For more simulation details, see (Wu et al., 2022b).**

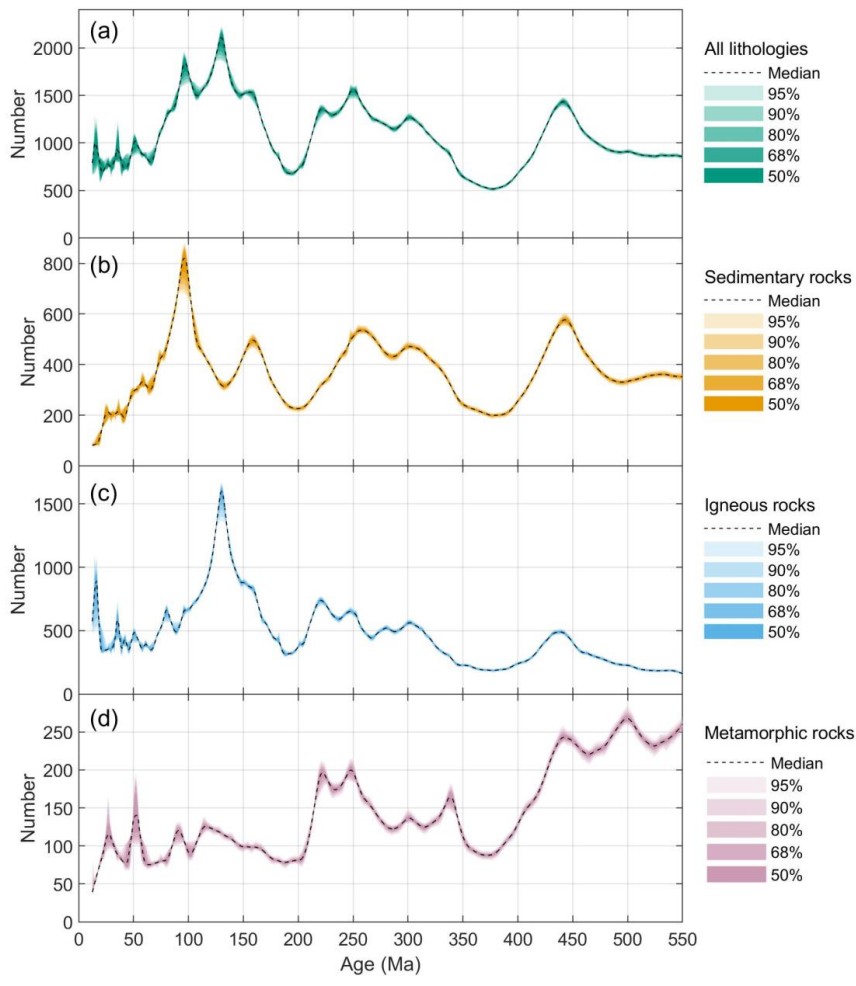


**Figure 9. Zircon production series in the Phanerozoic. The host rocks of each zircon series are (a) of all lithologies, (b) sedimentary rocks, (c) igneous rocks, and (d) metamorphic rocks. The filled zones represent 1000 Monte Carlo simulations of the zircon series with different transparencies, indicating different distribution probabilities, as shown in the legend. The dashed line indicates the median of the simulations. In each simulation, simulated zircon ages are selected based on their dating error. For more simulation**
**details, see (Wu et al., 2022b).**



### 3.4 Spatial characteristics of zircon production

At different geological times, the places where zircons grew in large quantities are also different. The spatial evolution of zircon production can be obtained by extracting the GPS of zircon sampling sites in certain age intervals. Although the present geographical locations differ from those of the past, this spatial distribution still has indicative significance. According to the zircon production peak periods given in Section 3.3, the spatial distribution of zircons for these periods can be plotted. Because age error varies with age, we use a 50-Myr age interval for zircon peaks in the Phanerozoic and a 100-Myr interval for the Precambrian data. Because of the similarity in the spatial distribution of zircons with similar ages, this paper only presents the spatial pattern of the main peak periods, i.e., $50 \pm 25$, $130 \pm 25$, $250 \pm 25$, $440 \pm 25$, $1000 \pm 50$, $1850 \pm 50$, $2500 \pm 50$, and $3400 \pm 50$ Ma (Figure 10). Inevitably, some areas were oversampled, such as China and Europe, and overly old ages are sparse due to preservation. In this case, we should pay more attention to relative rather than absolute zircon densities when comparing regions.

At $3400 \pm 50$ Ma in the Archean, zircons mainly grew in southern Africa, the southern Indian Peninsula, and the Himalayan Mountains (Figure 10h). At the Proterozoic-Archean boundary ($2500 \pm 50$ Ma), zircons mainly grew in the Rocky Mountains, northern North America, eastern South America, southern Africa, the Himalayas-Alps, and North China (Figure 10g). Compared to that of $2500 \pm 50$ Ma, the zircon growth area of the late Paleoproterozoic ($1850 \pm 50$ Ma) added northern Europe and the Alps (Figure 10f). In the early Neoproterozoic ($1000 \pm 50$ Ma), there were more areas where zircons grew in large quantities in eastern Africa and eastern Australia (Figure 10e). In the early Silurian ($440 \pm 25$ Ma), southern Africa and northern North America were no longer active in terms of zircon production (Figure 10d). In the Early Triassic ($250 \pm 25$ Ma), the Nordic region was no longer active, and zircons in Northeast China grew in large quantities (Figure 10c). In the Early Cretaceous ($130 \pm 25$ Ma), the Alps and Northwest China were no longer active (Figure 10b). In the early Cenozoic ($50 \pm 25$ Ma), eastern China was no longer active, and a large number of zircons grew mainly in the Rocky Mountains, the Andes, and the Himalayas to the Iranian Plateau (Figure 10a). For the spatial distribution of zircon ages of igneous, metamorphic, and sedimentary rocks, please see the supplementary materials (Figure S1-3).

## 4. Discussion

### 4.1 Recommended age

Generally, a recommended age is selected from the four zircon ages for geological interpretation. The empirical rule is that the $^{206}$Pb/$^{238}$U age is usually selected for <1.0 ($\pm 0.2$) Ga and the $^{207}$Pb/$^{206}$Pb age for > 1.0 ($\pm 0.2$) Ga. However, this rule of thumb is not always applicable and needs to be improved. Voice et al. (2011) suggested a threshold of ~1.2 Ga between $^{206}$Pb/$^{238}$U and $^{207}$Pb/$^{206}$Pb ages by analyzing ~38,000 zircon age data derived from LA-ICP–MS. On this basis, Spencer et al. (2016) added ~5000 SIMS zircon records and statistically proposed a cutoff age of ~1.5 Ga between $^{206}$Pb/$^{238}$U and $^{207}$Pb/$^{206}$Pb ages. Puetz et al. (2018) compiled ~420,000 zircon data, including LA-ICP–MS, SHRIMP, SIMS, and TIMS dating methods, using polynomial curve fitting models and obtained recommended ages as follows: $^{206}$Pb/$^{238}$U age for 0-1.2 Ga, $^{207}$Pb/$^{235}$U age for 1.2-2.0 Ga, and $^{207}$Pb/$^{206}$Pb age for > 2.0 Ga.

Fang et al. (2018) and Wu et al. (2019) proposed a new method of recommended age selection that involves analyzing zircon ages and errors from Chinese continental crust collected from the CNKI and Elsevier publishers, respectively. According to the decay principle of the U–Th–Pb isotope system and assuming that the influence of other factors can be ignored, the errors of $^{206}$Pb/$^{238}$U, $^{207}$Pb/$^{235}$U, and $^{208}$Pb/$^{232}$Th ages scale exponentially with age (see Table 3), where factors affecting the coefficients include but are not limited to isotope half-lives and experimental errors. The relationship between the $^{207}$Pb/$^{206}$Pb age error and age is more complex, so an empirical logarithmic formula obtained from a large number of experiments was selected for fitting (see Table 3), for which the fitting effect is good. Based on these error curves, the recommended age is the age type with the smallest dating error.

This study followed the selection criteria of Fang et al. (2018). After applying the age error fitting models to the zircon ages of the global continental crust, we suggest that the recommended ages for each age interval are 0-1163 Ma, $^{206}$Pb/$^{238}$U age; 1163-2390 Ma, $^{207}$Pb/$^{235}$U age; and > 2390 Ma, $^{207}$Pb/$^{206}$Pb age. This outcome is quite different from the empirical rule.

In addition, we also provide cutoff ages for different dating methods (see Table 4). When selecting the recommended age in the future, we should analyze the errors of various ages with caution instead of uncritically following the empirical rules.

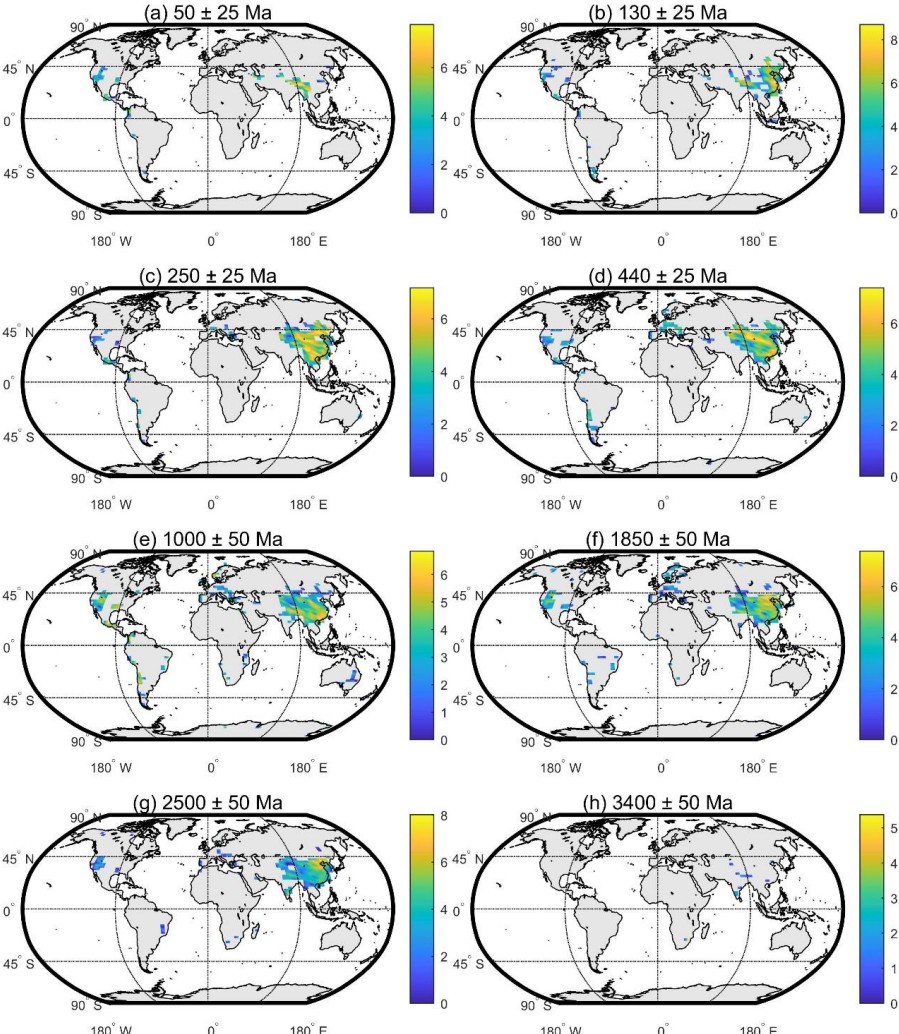

**Figure 10. Spatial distribution of zircon production peak periods of all lithologies. Since some areas were oversampled and old ages**
**are sparse, we should pay more attention to relative rather than absolute zircon densities when comparing regions.**

## 4.2 Temporal distribution

The amount of zircon production can indicate the intensity of geological activity (Hawkesworth et al., 2010). There are essentially two sources of zircon: magmatism and metamorphism (Hanchar and Hoskin, 2018). Igneous and metamorphic zircons can indicate the intensity of magmatic and metamorphic activities, respectively; they can also be combined to indicate
general geological activity. Detrital zircons are natural mixtures of igneous and metamorphic zircons. However, using zircon production as a proxy involves making certain assumptions, which can be weakened if the dataset used is large enough. Fang et al. (2018) and Wu et al. (2019) used independent zircon datasets from publishers CNKI and Elsevier to study the Chinese continental crust. The conclusions they obtained are surprisingly similar: some major orogenic movements in the Chinese continental crust agree with the peak periods of zircon production. Specifically, the zircon production peaks at 2500, 1850,



800, 440, 250, 130, and 50 Ma correspond to the Wutai Movement, Lvliang Movement, Jinning Movement, Caledonian Movement, Indo-China Movement, Yanshan Movement, and Himalayan Movement, respectively. This not only supports the reliability of big data statistics but also confirms that these geological events (periods of intense geological activity) are related to large amounts of zircon production.

Apart from internal dynamic events (such as orogeny and plate movement), zircon production series can also indicate
various associated surface processes. For example, enhanced climate denudation during orogeny may trigger crustal decompression melting and metamorphism (Yu et al., 2011; Tu et al., 2015). The melting of ice sheets can also lead to regional crustal uplift and related magmatic activities (Kim and Zhang, 2022; Mitrovica et al., 2001). Conversely, plate movement and volcanism will change the climate state by affecting the weathering process and atmospheric composition. Chemical weathering enhanced by continent-continent collisions lowers the $CO_2$ level of the atmosphere (Kidder and Worsley, 2004);
the formation of supercontinents is associated with increases in atmospheric oxygen (Campbell and Allen, 2008). In this framework, zircon production indicates the evolution not only of the crust but also of the Earth system.

Wu et al. (2020) further explored the indicative meaning of zircon production and provided a detailed evolution of Chinese continental crust by merging the zircon data of Chinese continental crust from publishers CNKI and Elsevier. The geological activities corresponding to the long-term peak periods of zircon production are more likely to be related to plate movement
(e.g., the assembly and breakup of supercontinents) and large-scale climate events (e.g., snowball Earth). The short-term zircon production peak periods may be affected by short-term tectonic and surface processes, such as crustal denudation and decompression melting caused by climatic factors.

The database presented in this paper integrates zircon dating data sampled from the global continental crust provided by multiple academic publishers, providing important clues for studying the evolution of the global continental crust and even
Earth's system. The zircon production peaks of the global continental crust are ca. 50, 130, 250, 300, and 440 Ma in the Phanerozoic (Figure 9) and ca. 800, 1000, 1850, 2500, 2700, 3200, and 3400 Ma in the Precambrian (Figure 8). The specific meaning of each peak period can be further explored. The temporal distribution of zircon production may vary by region, which can also be studied specifically. In short, this database is of great help for the future study of Earth system science, whether from a global scale or a regional scale and whether for Earth's entire history or a certain geological time.

### 4.3 Spatial distribution

The spatial distribution of zircon production peaks of all lithologies is presented in Figure 10. Obviously, at different geological times, the regional intensity of geological activity varied as well as the amount of zircon production. For the spatial evolution of detrital, igneous, and metamorphic zircons, please see the supplementary materials (Figure S1-3). Importantly, the zircon ages of some crustal regions show similar temporal evolution patterns, suggesting that these crustal regions might belong to
the same past tectonic unit or experienced similar geodynamic processes. Thus, we can conduct spatial classification by combining zircon temporal and spatial distribution by grouping regions with the same zircon temporal signature as a crustal unit. This classification allows us to obtain the spatial evolution of the continental crust.

The zircon database examined in this paper can provide powerful data support for the in-depth study of the temporal and spatial evolution of the continental crust. With this database, Fang et al. (2020) carried out a spatial classification of the Chinese
continental crust. Using the grid clustering algorithm, the zircon age series in different regions were compared, and the Chinese continental crust was divided into six crustal units, i.e., the Tibet-Sichuan-Yunnan, North Xinjiang, Northeast, Gansu-Qinghai, North China, and South China units. The zircon ages within each crustal unit have a similar time distribution. Intriguingly, these crustal units identified by zircons are basically consistent with those based on tectonics (Yang and Yu, 2015), verifying the scientific nature of zircon big data research and providing a new means of studying the spatial evolution of the crust.

In the future, this database can also be used to study the spatial distribution of zircons on a global scale or in various regions to obtain the spatial evolution of the continental crust. With the help of zircon spatial distribution, the spatial evolution of global tectonic zones can also be further studied to explore the formation and storage of oil, gas, and minerals, which will be of great help to commercial mining. Furthermore, the implications of zircon production for geodynamic processes may shed light on new energy sources.



### 4.4 Periodicity

Periodicity is obvious in the time series of zircon production. Wu et al. (2022b) used this database to systematically analyze the periodicity of zircon production in the global continental crust and obtained the following periods: (1) ca. 800, 360, 220, 160, 69, 57, 44, 30, 20, and 17 Myr for zircons from all lithologies; (2) ca. 680, 290, 160, 100, 45, 29, 24, 20, and 17 Myr for igneous zircons; and (3) ca. 680, 150, 100, 67, 56, 44, 31, 28, and 24 Myr for metamorphic zircons. These results are reliable since various time series analysis methods were applied, including the MTM, periodogram, wavelet transform, and evolutionary power spectrum methods. In addition, Monte Carlo simulation was used to evaluate the impact of dating errors. Other zircon databases were also studied to obtain zircon production cycles. Prokoph and Puetz (2015) gave zircon cycles of ca. 2300, 1600, 800, 550, 280, 230, 180, and 60 Myr using the zircon dataset from Condie (2013) and the wavelet transform method. Puetz et al. (2018) and Puetz and Condie (2019) analyzed the periodic harmonics of ~800, 270, 180, and 90 Myr using the periodogram method. The results of these studies are basically consistent despite showing some divergence, confirming the reliability of zircon periodicity.

Beyond zircon production cycles, we can explore the geodynamic mechanism and driving source. The zircon production cycles are not only consistent with the periodicity of various geological events but also agree with many astronomical cycles, suggesting a possible link between the astronomical environment and geological processes. Specifically, the ca. 800 Myr period of zircon production corresponds to the supercontinent cycle (Li and Zhong, 2009; Nance et al., 2014) as well as the precession period of the galactic warp (~ 600 Myr) (Poggio et al., 2020). Many geological events have a period of ~220 Myr, such as magmatic activity (Isley and Abbott, 2002), sea level changes (Boulila et al., 2018), biological extinctions, and geomagnetic reversals (Rampino and Stothers, 1984b). This ~220-Myr period is consistent with the galactic year (the revolution period of the sun around the galactic center) (Bland-Hawthorn and Gerhard, 2016). The geological events with a 160-Myr period include a long-term glacial cycle (~150 Myr) (Chumakov, 2002, 2005), fossil diversity (Rohde and Muller, 2005), paleoclimate change (Prokoph et al., 2008; Veizer et al., 2000), and large igneous provinces (LIPs) (Prokoph et al., 2013). Correspondingly, the radial movement period of the solar system relative to the galactic center is ca. 149 Myr (Gardner et al., 2011; Schönrich et al., 2010). Shaviv (2003), however, reported another related astronomical period of $143 \pm 10$ Myr, namely, the period of the solar system crossing the spiral arms of the Milky Way. The period of ~60 Myr presents geological features such as fossil diversity (Rohde and Muller, 2005; Roberts and Mannion, 2019), LIPs (Prokoph et al., 2013), depositional cycles (Meyers and Peters, 2011), and geomagnetic reversals (Negi and Tiwari, 1983) corresponding to the vertical motion period of the solar system relative to the galactic disk (Randall and Reece, 2014). The ~30-Myr cycle has been extensively explored and is reflected in many geological features, including biodiversity (Roberts and Mannion, 2019; Raup and Sepkoski Jr, 1984), LIPs (Prokoph et al., 2013), geomagnetic reversals (Melott et al., 2018; Stothers, 1986), subduction zone migration (Müller and Dutkiewicz, 2018), climate change (Boulila, 2019; Boulila et al., 2021), and meteorite impacts (Rampino and Stothers, 1984a, b). Astronomically, the ~30-Myr cycle is often related to the period of the sun's transit of the galactic disk (half the period of the vertical motion) (Randall and Reece, 2014).

Zircon periodicity can be further explored, for which the zircon database undoubtedly provides excellent research materials. On the one hand, it is possible to deeply explore the source of Earth's dynamics and the underlying mechanisms involved. Although many researchers believe that the driver of Earth's processes comes from Earth's interior (Mitchell et al., 2022), increasing evidence shows that astronomical factors can also have various and multiscale influences on the evolution of Earth's system, from short-term climate change to long-term plate tectonics (Hays et al., 1976; Zaccagnino et al., 2020; Carcaterra and Doglioni, 2018). Unlike the widely accepted Milankovitch theory, which explains the astronomical influence on Earth's climate (Milankovitch, 1941), it is still under debate whether and how the astronomical environment will affect long-term geological activities. Fortunately, the zircon database used in this study provides excellent materials for research on geodynamics and the mechanisms behind them, which is essential to verifying or falsifying the astronomical influence hypothesis. On the other hand, in the interdisciplinary subject of astrogeology, astronomical parameters must be improved in both precision and accuracy if used in Earth system science. For example, the vertical motion period of the solar system relative to the galactic disk is still under debate; is it ~90 Myr or ~60 Myr? (Kramer and Randall, 2016; Schutz et al., 2018; Randall and Reece, 2014). This uncertainty is impactful for Earth system science but negligible for astronomy. Once the astronomical impact mechanism is available, the zircon database will provide good constraints for astronomical parameters and their



evolution over time for deep-time research.

**4.5 Hot data issues**

When using the number of zircon ages to indicate the intensity of geological activities, two conditions must be met: the random exposure of the crust and the random sampling of zircons. However, sampling bias, which occurs when zircons are oversampled from a certain area or time interval, is inevitable and referred to as a hot data issue (Wu et al., 2020; Puetz et al., 2017; Puetz et al., 2018). In addition, considering the convenience and feasibility of fieldwork, sampling sites must be places that humans can reach. As shown in Figure 1, the sampling sites are dense in Europe and China but sparse in the Sahara and Siberia. Since zircon production is inherently unevenly distributed in the crust, the identification of hot data is more complicated. After all, it is difficult to determine whether the large amount of zircon data is caused by artificial oversampling or crustal conditions that are suitable for zircon growth.

To solve hot data issues, Puetz et al. (2017) proposed the methods of grid-area and modern-sediment sampling using the surface area to weigh the zircon data. However, this approach is more suitable for studying the exposed crust than it is for studying the evolution of the crust (Wu et al., 2022b). Some important magmatic and metamorphic activities in crustal evolution (such as orogenic belts and subduction zones) were concentrated in certain regions and periods, so the distribution of zircon ages was uneven by nature. Alternatively, Wu et al. (2020) proposed the W index and Y index to measure the impact of hot data. The W index should be used when artificial oversampling leads to a visual peak of zircon production without geological meaning. We can obtain the W index by comparing the span of the zircon peak with the dating error. If the peak span is even smaller than the dating error, then the peak is probably caused by artificial oversampling (hot data). Conversely, if artificial oversampling complements the trough period of zircon production, rendering the original zircon peak no longer prominent, zircon production will display a "homogeneous" trend. In this case, we can use the Y index, the aspect ratio of a zircon peak, to evaluate this "homogeneous" effect. If the zircon peak is too broad (the Y index is too large), it might be affected by the "homogenization" effect (another biased sampling). However, with the help of the W index and Y index, we can only measure the impact of biased sampling and cannot address the root of the problem.

Instead, Wu et al. (2022b) proposed eliminating the influence of hot data from the result based on coherence. Given a research area, one can label part of the area that tends to contain hot data as region 1 and the remaining area as region 2. Then, one can calculate the coherence between the time series from regions 1 and 2. The reasoning is that the frequency signal caused by the hot data should only exist in the time series of region 1, suggesting little similarity between the two series in the frequency band contaminated by hot data, i.e., the degree of coherence is low, while the frequency band with high coherence indicates high similarity between regions 1 and 2 and reflects the inherent properties of zircon production. In this way, we can not only filter out the influence of hot data but also retain the original geological information of biased-sampled regions. However, this method cannot quantitatively identify regions that tend to contain hot data and needs to be further improved.

**5. Data availability**

The database described in this manuscript can be accessed at Zenodo repository at https://doi.org/10.5281/zenodo.7387567 (Wu et al., 2022a).

**6. Conclusions**

Here, we introduce the largest known zircon U–Th–Pb geochronology database of the global continental crust. This database provides comprehensive research materials for Earth system science due to its large amount of data (~2 million records), wide sampling range (global continental crust), comprehensive samples (detrital, igneous, and metamorphic zircons), and various dating methods (LA-ICP–MS, SHRIMP, SIMS, TIMS, etc.).

Based on this database, we described the characteristics of zircon dating errors, compared different dating methods, and discussed hot data issues and possible solutions. When selecting recommended zircon ages, empirical rules should be used



critically. By analyzing the dating errors of various ages, we recommend using $^{206}Pb/^{238}U$ age for 0 – 1163 Ma, $^{207}Pb/^{235}U$ age for 1163 – 2390 Ma, and $^{207}Pb/^{206}Pb$ age for > 2390 Ma to reduce the uncertainty of the recommended ages. Since the

recommended age intervals vary with the dating method, we suggest that the selection of recommended ages should consider the data used in specific research. In addition, sampling bias would affect the objectivity of statistics to some degree. Although at the present stage we can only evaluate this effect from results, the rich spatiotemporal information of this database provides a good experimental platform for exploring potential solutions to the root causes of problems.

This zircon database provides excellent materials for multiple fields of research, including but not limited to crustal

growth and evolution, supercontinent cycles, plate tectonics, paleoclimate changes, and biological extinction. The amount of zircon production can indicate the intensity of geological activities and be used to study the evolution of the continental crust and Earth system, whether from the global or regional scale and whether for Earth's entire history or a specific period. Combined with other geological processes and the astronomical environment, we can also use the frequency signals of the zircon age series to explore the geodynamic mechanisms and their potential drivers. In addition, this database has potential

applications in the commercial mining of oil, gas, and minerals if associating structural geology with the temporal and spatial distribution of zircon production.

## Supplement

## Author contributions

YW, XF, and JJ compiled the data.
YW and JJ merged the data, formatted the data, performed the analyses, standardized the reference materials, organized the database, managed the publication of the database on the Zenodo repository, and drafted and revised the manuscript.
JJ initiated and supported data compilation.

## Competing interests

The author for correspondence declares that neither they nor their coauthors have any competing interests to report.

## Acknowledgments

We would like to thank the following people for their help in collecting the data: Muyuan Zhu, Sisi Liao, Lizhi Xue, Zhe Chen, Jiangnan Yang, Yamin Lu, Kun Ling, Shengyi Hu, Shuyuan Kong, Yiwei Xiong, Huacheng Li, Xiuqi Shang, Rui Ji, Xueyun Lu, Biao Song, and Lei Zhang.

**Financial support**

None.

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
