# Peer review of "A global zircon U-Th-Pb geochronological database"

_Earth System Science Data, 2023_

## Author Response (AR1)

**Point-by-point response for revised submission**

A global zircon U–Th–Pb geochronological database

Yujing Wu et al.

*Earth Syst. Sci. Data Discuss.*, https://doi.org/10.5194/essd-2023-20

RC: Referee Comment, AR: Author Response, AC: Author Change

Dear Editor,

Thank you very much for your precious time and effort spent on our manuscript and dataset. The referees' comments and suggestions were very helpful. We did a major revision on our manuscript and updated our zircon database (version v2) in the Zenodo repository to improve our work and address referees' concerns. The modification work mainly involves four aspects.

1) Supplement data in the database.

2) Recalculate the time series and update corresponding figures.

3) Abridge out-of-scope contents and supplement data description contents.

4) Polish the English language of the entire revised manuscript.

There was a small portion of the modifications not fully complying with the referees' advice, but we explained the reasons in detail. Please find below a point-by-point reply to all referee comments and the corresponding revisions.

Many thanks.

Kind regards,

Yujing Wu (on behalf of the author team)

**Author Response to Referee #1**

A global zircon U–Th–Pb geochronological database

Yujing Wu et al.

*Earth Syst. Sci. Data Discuss.*, https://doi.org/10.5194/essd-2023-20

RC: Referee Comment, AR: Author Response, AC: Author Change

Dear referee,

Thank you very much for your positive response, and for your precious time and effort spent reviewing the manuscript and the dataset. Your suggestions are of great help. Please find a point-by-point reply below.

Kind regards,
Yujing Wu (on behalf of the author team)

**Comments and responses:**

**RC: General Comments:**

The U-Pb database discussed in this manuscript is certainly publishable, and it could provide the research community with valuable. However, at this point, I'm still unsure about its actual value for multiple reasons. Despite these concerns, I recommend publishing the database, but only after major revisions to the manuscript. The areas of concern are threefold: (a) using outdated and inaccurate methods for determining the best U-Pb age, which is more directly related to the degree of concordance than it is to the age-uncertainty, (b) concerns about the percentage of ages that have null values, these should be stated in the revised manuscript, and (c) significant grammar errors and/or poor word choices that will likely require a reliable editing service to correct. Further details related to these items follow.

**AR: General Responses:**

Thank you very much for your comments. We will revise the manuscript as you suggested. The following is our reply to your concerns.

(a) We will reselect the best U-Pb ages using the new non-iterative probability method you suggested and discuss its advantages. But, we would like to keep the original age series in the main text or supplementary materials for readers to compare.

(b) We will add the $^{206}Pb/^{238}U$, $^{207}Pb/^{235}U$, and $^{207}Pb/^{206}Pb$ ratios and uncertainties to the Zenodo repository and give detailed statements of the null values in the revised manuscript.

(c) We feel so sorry for our poor English expression. Thank you very much for your kind suggestions on the grammar and word choices. We will find a more reliable editing service for the revised manuscript.

**AC: General Changes:**

Thank you again for your comments. We have revised the manuscript as you suggested. The following are our general revisions.

(a) We reselected the recommended U-Pb ages using the non-iterative probability method you suggested. The original results were moved to the supplementary materials.

(b) We added the $^{206}Pb/^{238}U$, $^{207}Pb/^{235}U$, and $^{207}Pb/^{206}Pb$ ratios and uncertainties to the Zenodo repository (see version v2: https://doi.org/10.5281/zenodo.8040079) and give detailed statements of the null values in the revised manuscript.

(c) We purchased a Gold Language Editing service from Springer Nature to polish the entire revised manuscript. We hope it reads better now.

**RC: Specific Comments:**

RC: Table at line 118: The database lacks key details, such as the $^{206}Pb/^{238}U$, $^{207}Pb/^{235}U$, and $^{207}Pb/^{206}Pb$ ratios and uncertainties, the depositional/stratigraphic ages, and many records have ages and GPS coordinates that are missing. Despite these deficiencies, the database still has considerable potential for solving outstanding geological problems, but less so than if all data items were completed. The authors should mention the percentage of records that have null values.

AR: We appreciate very much your affirmation of our work. First, we did collect the $^{206}Pb/^{238}U$, $^{207}Pb/^{235}U$, and $^{207}Pb/^{206}Pb$ ratios and uncertainties. We will add these data to the Zenodo repository. Second, for the missing ages and GPS coordinates, we will state the percent of the null values in detail in the revised manuscript. We wish we could fill these values but the original papers publishing the zircon records don't include them. We have to leave these items empty for the sake of authenticity.

AC: We have supplemented the $^{206}Pb/^{238}U$, $^{207}Pb/^{235}U$, $^{207}Pb/^{206}Pb$, and $^{208}Pb/^{232}Th$ ratios and uncertainties in the zircon database. Please see the updated database (version v2: https://doi.org/10.5281/zenodo.8040079) in the Zenodo repository for more details.

RC: Lines 156-157: This sentence currently states: "Although TIMS is more precise, other methods are more efficient and widely used (Gehrels, 2014)." This is not the exact reason. Perhaps rephrase this as: "Although TIMS is more precise, methods such as LA-ICP-MS are more cost effective and thus are more widely used (Gehrels, 2014)."

AR: Will be implemented. Thanks for the suggestion.

AC: Implemented. Please see lines 323-324 in the revised manuscript with track-changes.

RC: Line 161-164: Using an arbitrary cutoff-age at 1000 Ma to select the best U-Pb age, as proposed in this manuscript, is flawed. Instead, Puetz et al. (2021) and Puetz & Spencer (2023) published a non-iterative probability method that (a) eliminates the artificial depression in the U-Pb age distribution at 1000 Ma caused by the

arbitrary cutoff method, and (b) produces consistent age-distribution based on the degree of discordance without producing the artificial depression at 1000 Ma. It is suggested that the authors review these papers and mention these advantages, as discussed in detail in the references below:

Puetz, SJ; Spencer, CJ; Ganade, CE (2021). Analyses from a validated global U-Pb detrital zircon database: Enhanced methods for filtering discordant U-Pb zircon analyses and optimizing crystallization age estimates. Earth-Science Reviews 220, 103745. https://doi.org/10.1016/j.earscirev.2021.103745

Puetz, SJ; Spencer, CJ (2023). Evaluating U-Pb accuracy and precision by comparing zircon ages from 12 standards using TIMS and LA-ICP-MS methods. Geosystems and Geoenvironment 2, 100177. http://dx.doi.org/10.1016/j.geogeo.2022.100177

AR: We appreciate you very much for your references. We have carefully read these two papers and will discuss the non-iterative probability method in the revised manuscript. Although it doesn't matter anymore, we used cutoff ages as shown in Table 4 rather than 1000 Ma.

AC: We recalculated the zircon age series with recommended ages derived by the non-iterative probability method. Accordingly, we updated the figures and some sentences. Please see the new Figures 4 and 5 and lines 396-397 in the revised manuscript with track-changes. To avoid redundancy and misunderstanding, we also moved the tables and figures related to cut-off ages to the supplementary materials for readers' information (Figures S1-4 and Tables S1-3).

RC: Line 183-185: Regarding the sentence: "Therefore, the amount of zircon production can be used to understand the past intensity of geological activity (Hawkesworth et al., 2010)." … Hawkesworth et al. (2010) is a poor reference to support this statement. Instead, the following reference is suggested:

Arndt, N; Davaille, A (2013). Episodic Earth evolution. Tectonophysics 609, 661-674. https://doi.org/10.1016/j.tecto.2013.07.002

AR: Thanks for the reference. We will revise it as you suggested.

AC: Revised. Please see lines 393-394 in the revised manuscript with track-changes.

RC: Lines 185-194: These lines discuss the results in Figures 8 and 9. Importantly, the age distributions in these figures are raw age counts. The usage of raw age counts is less than optimal because it favors age peaks in heavy sampled regions while failing to show significant age peaks in sparsely sampled regions. For instance, age distributions from a database heavily populated with samples from China, as the database here has, will show strong age-peaks at 800 Ma and 2500 Ma. However, another database with minimal samples from China will tend to show a weak peak at 800 Ma, and a peak at 2700 Ma that is far stronger than the 2500 Ma peak. One way around this problem of disproportionate sampling is to weight the records inversely proportionally to the sampling densities. Then, the resulting age distributions will be remarkably consistent, despite the divergent sampling densities for each database. This suggestion is easy to test simply by first

weighting the records inversely proportional to sampling densities, and then summing the age-counts by using the weights. For details about this method, refer to Puetz et al. (2017), *Quantifying the evolution of the continental and oceanic crust,* which is already in the reference list.

AR: Thank you very much for your suggestion. We will calculate new zircon production series by adding weights in the revised manuscript. However, we want to keep the original series using raw age counts in the manuscript. In this way, readers can have more a direct understanding of our database and compare the differences brought about by the weights.

AC: We did recalculate the zircon production series by adding weights according to sampling densities. Instead of weighing the records inversely proportional to spatial sampling densities as you suggested, we applied inverse proximity weighting referring to Mehra et al. (2021) (https://doi.org/10.1130/GSATG484A.1), which considers both spatial and temporal sampling density. The basic ideas are similar. Please see lines 397-399 and the new Figures 4 and 5 in the revised manuscript with track-changes, and the Methods section and Figure S5 in the revised supplementary materials.

RC: Line 212: Regarding the sentence: "At different geological times, the places where zircons grew in large quantities are also different." This is already well known and is commonly referred to as the globally heterogeneous distribution of magmatic ages (Hawkesworth et al., 2010; Puetz et al., 2017; and many others). Suggest replacing this sentence by stating that the database here supports the globally heterogeneous distribution of magmatic U-Pb ages.

AR: Will be revised as you suggested.

AC: Revised. Please see lines 410-411 in the revised manuscript with track-changes.

RC: Line: 220: The authors propose a very subjective approach, with no details on how to accomplish adjustments for the different regional sampling densities. As already explained in the comments related to lines 185-194, the simple and standard approach to solving this problem is to weight the records inversely proportional to sampling densities.

AR: We will weigh the records inversely proportional to sampling densities in the revised manuscript.

AC: In the revised manuscript, we applied the inverse proximity weighting method to calculate the zircon age series referring to Mehra et al. (2021) (https://doi.org/10.1130/GSATG484A.1). Both temporal and spatial sampling densities were considered. Please see lines 397-399 and the new Figures 4 and 5 in the revised manuscript with track-changes, and the Methods section and Figure S5 in the revised supplementary materials.

RC: Lines 252-254: The method that the authors propose here is seriously flawed, based on tests in Puetz et al. (2021) and Puetz & Spencer (2023) – which compared highly accurate and precise TIMS ages with LA-ICP-MS ages. Using $^{206}Pb/^{238}U$

ages for 0-1163 Ma; $^{207}Pb/^{235}U$ ages for 1163-2390 Ma, and $^{207}Pb/^{206}Pb$ ages when > 2390 Ma is a flawed system. Specifically, the magnitude of the uncertainty (the imprecision) is not directly related to the accuracy of the age. Read Puetz et al. (2021) and Puetz & Spencer (2023) for details about this method. Studies in those papers show that the best U-Pb age gradually transitions from the $^{206}Pb/^{238}U$ age at ~400 Ma to the $^{207}Pb/^{206}Pb$ age at ~1600 Ma. Between those points, the best age gradually transitions from ~400 Ma to ~1600 Ma based on a non-iterative probability model.

AR: Thank you for this new method. We will use this non-iterative probability model to recalculate our data. Again, we want to keep the original series either in the main text or in the supplementary materials for readers to compare differences. Another reason is that we are not sure how much our series are influenced by the cutoff ages since we applied various bin sizes and Monte Carlo simulation to minimize the influence of age uncertainty, which were not used in the papers you suggested. Presenting the results calculated by two methods is a good opportunity to compare and test.

AC: We recalculated our time series using the non-iterative probability model you suggested. Please see lines 396-397 and the new Figures 4 and 5 in the revised manuscript with track-changes. In addition, we moved the original series in the supplementary materials (Figures S6 and S7).

RC: Line 290: Once again, the statement that "The zircon production peaks of the global continental crust are…" is biased by using raw age counts rather than weighting the records inversely proportional to sampling densities.

AR: We will weigh the records inversely proportional to sampling densities in the revised manuscript.

AC: We weighted the dating records according to the inverse proximity weighting method in Mehra et al. (2021) (https://doi.org/10.1130/GSATG484A.1). Both temporal and spatial sampling densities were considered. Please see lines 397-399 and the new Figures 4 and 5 in the revised manuscript with track-changes, and the Methods section and Figure S5 in the revised supplementary materials.

RC: Lines 303-309: This regionally based approach is good, and in this instance, does not necessarily require weighting the records inversely proportional to sampling densities.

AR: Thank you for your support.

AC: Thanks again.

RC: Lines 316-319: The age distributions (and thus the periodicities) for detrital, igneous, and metamorphic samples should be nearly identical. Again, if the authors recalculate the age distributions by weighting the records inversely proportional to sampling densities, then I suspect the periodicities will be essentially the same. Inaccurate age-distributions will produce incorrect periodicities. Another important requisite for testing periodicity is to de-trend the data. My question to

the authors: Were the age-distributions de-trended prior to spectral analysis?

AR: First, we will weigh the records inversely proportional to sampling densities in the revised manuscript. Second, the age distributions were de-trended before spectral analysis.

AC: We weighted the dating records according to the inverse proximity weighting method in Mehra et al. (2021) (https://doi.org/10.1130/GSATG484A.1). Both temporal and spatial sampling densities were considered. Please see lines 397-399 and the new Figures 4 and 5 in the revised manuscript with track-changes, and the Methods section and Figure S5 in the revised supplementary materials.

RC: Lines 325-347: These are interesting studies that require more rigorous analyses to determine their reliability.

AR: Great point! In this data description paper, we tend to introduce more potential research values of this database. These intriguing but controversial studies might be better verified in the future using this zircon database as one of the supporting materials.

AC: We finally removed the sentences on these controversial studies, which were beyond the scope of a data description paper for the ESSD journal.

RC: Lines 373-374: Regarding the sentences: "To solve hot data issues, Puetz et al. (2017) proposed the methods of grid-area and modern-sediment sampling using the surface area to weigh the zircon data. However, this approach is more suitable for studying the exposed crust than it is for studying the evolution of the crust." … This statement is false and it is suggested that it be removed. Weighting records inversely proportional to sampling densities is a STANDARD approach (refer to references in Puetz et al., 2017). However, If the authors actually believe this statement is true, then the authors should present the test that they used to demonstrate this. However, I suspect this is an unsupported statement. For instance, numerous studies over the past 50 years have shown that the age distributions are remarkably similar regardless of depth or height. Parman (2015) shows similar findings – the age distributions remain remarkably similar over time (each involving samples of different depths).

AR: Thanks. We will revise or remove this statement as you suggested.

AC: We removed the inappropriate statement, added a sentence to explain the age distributions, and cited the paper of Parman (2015). Please see lines 595-597 in the revised manuscript with track-changes.

**Grammar related items:**

RC: Line 11: Grammar error / typo. Delete the words "and theses"

AR: The words "and theses" will be replaced by "and dissertations". Sorry for the confusion.

AC: We replaced "and theses" with "and dissertations". Please see line 12 in the revised manuscript with track-changes.

RC: Line 14: Poor word choice. Suggest replacing the "weaken" with "minimize"

AR: Will be implemented. Thank you.

AC: We replaced "weaken" with "minimize". Please see line 14 in the revised manuscript with track-changes.

RC: Line 17: Instead of mining and energy, is the intent to state: "mining and energy exploration"?

AR: Yes, correct. Will be revised as you suggested.

AC: We added the word "exploration" there. Please see line 17 in the revised manuscript with track-changes.

RC: Line 45-46:   As it is currently written, this sentence does not make sense, and in fact, is false: "However, in most cases, these zircon samples were used for independent regional studies and would probably not be used thereafter (Wu et al., 2019)." … In fact, over the past 20 years, numerous authors have re-used the data for these regional studies for further regional analyses and well as in global compilations for global analyses.

AR: This wrong sentence will be deleted.

AC: Deleted.

RC: Lines 46-51: These lines should be deleted and rephrased in one sentence to state something like the following: Here, we expand upon previous global databases of U-Pb dated zircon, which could provide a means for enhanced academic and commercial geological analyses.

AR: Will be implemented.

AC: We deleted the previous lines 46-51 and rephrase them as: Here, we collected zircon U-Th-Pb dating records for the past decades and built a global zircon database, which could provide a means for enhanced academic and commercial geological analyses. Please see lines 66-67 in the revised manuscript with track-changes.

RC: Line 62: Suggest deleted the unnecessary words at the end of this sentence "in which the earth is located"

AR: Will be implemented.

AC: We deleted "in which the earth is located".

RC: Line 63-64 currently state: "However, if the amount of data is not sufficient, the resolution of zircon age series will be lower, leading to possible analysis bias. In addition, limited sampling locations will also affect the objectivity of statistics." This sentence is too wordy and confusing. Thus, suggest making this clearer by simply stating something like the following: "Insufficient data with limited global coverage can affect results, which in turn can contribute to misleading interpretations."

AR: Will be implemented.

AC: The sentence was revised as you suggested. Please see lines 79-80 in the revised manuscript with track-changes.

RC: Line 67-68: Suggest deleting this sentence: "Undoubtedly, this database provides a more comprehensive and objective chronology data source on both the time and space dimensions for future earth system science research." This interpretation is too strongly worded and even questionable. Only further independent studies (from research teams other than the current set of authors) will determine the usefulness of this global database.

AR: Will be implemented.

AC: We changed "undoubtedly" with "may", and rephrased the sentence accordingly. Please see line 82 in the revised manuscript with track-changes.

RC: Line 70: Suggest revising "other geological events and astronomical environments" to state "other geological and astronomical events"

AR: Will be implemented.

AC: Revised as you suggested. Please see line 86 in the revised manuscript with track-changes.

RC: From this point forward, I will no longer make suggestions related to grammar and interpretations. Even while the manuscript is generally understandable, it is riddled with grammar errors, poorly phrased sentences, and awkwardly phrased sentences. Therefore, it is suggested that the authors find a proficient proofreader or editing service to revise the entire manuscript to conform to standard English grammar and phrasing of words.

AR: Sorry for the inconvenience. We will find a more advanced English editing service for the revised manuscript.

AC: We purchased a Gold Language Editing service from Springer Nature to revise the entire manuscript. Hope the revised manuscript reads better.

RC: Line 268: Again, use the word "minimized" rather than "weakened"

AR: Will be implemented.

AC: Revised as you suggested. Please see line 529 in the revised manuscript with track-changes.

**Author Response to Referee #2**

A global zircon U–Th–Pb geochronological database

Yujing Wu et al.

*Earth Syst. Sci. Data Discuss.*, https://doi.org/10.5194/essd-2023-20

RC: Referee Comment, AR: Author Response, AC: Author Change

Dear referee,

Thank you sincerely for your response, and for dedicating your valuable time and effort to reviewing both the manuscript and the dataset. We deeply appreciate your insightful advice and concerns. We will meticulously address each of your points in the revised manuscript and updated database. Please find our point-by-point reply below.

Kind regards,
Yujing Wu (on behalf of the author team)

**Comments and responses:**

**RC: General Comments:**

Wu et al. describe an updated compilation of zircon U-Th-Pb ages from journal articles and dissertations. A previous compilation focussing on Chinese geochronology (Wu et al., 2019) has been expanded to include additional samples from across the world. A recent publication in Earth-Science Reviews (Wu et al., 2022) also describes this database. The dataset is published as two excel spreadsheets on Zenodo, with a third document containing the source references.

Whilst the compilation of data from ~12,000 papers is a commendable effort that could support diverse future research, the database presented here lacks important additional information that would allow quality assessment and control, such as more details on the analytical method and age correction. The original data sources should be included in the manuscript reference list. Beyond the description of the dataset, the manuscript further contains scientific interpretations and discussions that go beyond the scope of Earth System Science Data and would require rigorous, additional scientific review.

I cannot recommend this manuscript for publication in its present form due to several concerns detailed below. I would be willing to review the data description again if these concerns can be addressed, however, I recommend the scientific discussion (Sections 3 to 4.4) be removed from the manuscript.

**AR: General Responses:**

We greatly appreciate your comments and feedback. We will carefully revise the manuscript and make the necessary updates to the database to address your

concerns. Please find below our general response to your concerns.

Firstly, to ensure quality assessment and control, we will enhance our Zenodo repository by including zircon reference materials such as 91500 and GJ-1, which were used for age correction. Additionally, we will provide isotopic ratios ($^{206}Pb/^{238}U$, $^{207}Pb/^{235}U$, $^{207}Pb/^{206}Pb$, and $^{208}Pb/^{232}Th$) along with their associated uncertainties. By doing so, readers can use this zircon database to calculate new ages as needed. In addition, it is important to acknowledge that null values may exist due to the absence of provided information in the original literature. Despite these null values, the remaining data remains useful. In the revised manuscript, we will include detailed statements regarding the presence of null values to ensure transparency and clarity for readers.

Secondly, we would like to clarify that we prefer to include the source references in the Zenodo repository rather than in the manuscript itself. Because the original data sources encompass more than 10,000 papers, and including them in the manuscript's reference list would extend it to over 400 pages. By placing the source references in the Zenodo repository, we can still provide readers with access to the references and in the meantime respect previous academic achievements.

Thirdly, we are willing to make some reductions in the out-of-scope scientific interpretations and discussions. Our initial goal was to introduce the characteristics and potential value of our database, but we may have provided excessive detail. To address this, we will remove Sections 4.1 to 4.4 and instead summarize the content in one or two paragraphs. Additionally, we will make efforts to condense Section 3 as much as possible to ensure that data description is the main purpose of the paper. However, we still want to retain an abridged version of Section 3 in the manuscript since it presents the fundamental characteristics of the zircon data clearly and intuitively.

**AC: General Changes:**

Thank you again for your comments and feedback. We have carefully revised the manuscript and made the necessary updates to the database to address your concerns.

Firstly, we added the following contents to the updated database (version v2: https://doi.org/10.5281/zenodo.8040079): isotopic ratios ($^{206}Pb/^{238}U$, $^{207}Pb/^{235}U$, $^{207}Pb/^{206}Pb$, and $^{208}Pb/^{232}Th$) and uncertainties, and reference materials which were used to derive zircon ages. We also added the proportion of non-null values for each data field in Table 1, along with corresponding statements in the manuscript.

Secondly, we supplemented some literature information in the "References" file, including the literature title, first page, last page, DOI, and URL (see the updated database version v2: https://doi.org/10.5281/zenodo.8040079). However, we keep this "References" file in the Zenodo repository to avoid a lengthy reference list. This way provides readers with access to the original references and in the meantime respects previous academic achievements.

Thirdly, we did significant abridgment to the out-of-scope contents. We shorten the original sub-sections 4.1 to 4.4 into a new concise sub-section 5.1. We

also condensed the original section 3 into a new short section 4 with only three paragraphs.

**RC: Specific Comments:**

RC: It is unclear how much of the presented data compilation was already included in Wu et al. (2019) and Wu et al. (2022) vs. what has been added since. What is the added value of this present database that it should not just be an updated version of the previous publications?

AR: Thank you for your comments. We will detail the parts of the database used in previous publications and clarify the newly added components in the revised manuscript. Since previous publications were conducted during the data compilation without disclosing the data, we treat this database as the initial publicly available version, rather than an update.

AC: There were ~0.41 million and ~2 million dating records used by Wu et al. (2019) and Wu et al. (2022), respectively. However, none of the previous papers disclosed the data. Please see lines 94-101 in the revised manuscript with track-changes for the details of the zircon data used in previous studies.

RC: Quality assessment/quality control: there is very little description of the curatorial procedure during compilation of the dataset; e.g. information on the recalculation of uncertainties (if any) where sources are inconsistent or on how lithologies were assigned (curatorial decision or is this information contained within the data sources?). Furthermore, very little metadata is provided that would allow others (including myself) to assess data quality.

AR: Great point! In the revised manuscript, we will include a dedicated section that provides a detailed overview of the curatorial procedure. Please find below the response to the questions you listed:

Firstly, we did recalculate some uncertainties to normalize the errors to standard deviation. The original references contained uncertainties in various forms, such as relative uncertainty and 2 standard deviations. We have developed specific processing methods to handle these different forms, and we will provide comprehensive details on these methods in the revised manuscript.

Secondly, we categorized the lithology into three groups (sedimentary, igneous, and metamorphic) based on the information from the original literature. In instances where the data sources did not provide any lithology information, we left the "Lithology" field empty. We will explicitly state the proportion of null values in the revised main text.

Thirdly, regarding quality control, we have checked twice to ensure that the collected data was consistent with the information provided in the original literature during the database construction. The zircon data were directly collected from the original literature, and we made no changes to the data itself, only standardized the forms to preserve the original content as much as possible. We are confident about the authenticity and consistency of our data, and welcome others to do quality check by comparing the data in the database with the data in

the original reference. The information of the references is provided in the Zenodo repository. However, the quality of the geochronology data itself (e.g. how age correction was applied and how uncertainty was derived) depends on the original references. Interested researchers can conduct in-depth studies based on the reference file we provided. As mentioned earlier, we will update our database to include isotopic ratios and uncertainties (including $^{206}Pb/^{238}U$, $^{207}Pb/^{235}U$, $^{207}Pb/^{206}Pb$, and $^{208}Pb/^{232}Th$), as well as zircon reference materials (such as 91500 and GJ-1) for age correction. This will enable researchers to have the option to recalculate zircon ages or do necessary corrections in their own way instead of relying solely on the ages provided in the original literature. This aligns with our goal of providing researchers with a comprehensive data compilation for further investigation.

AC: We added a new section 3 on data cleaning, including procedures about uncertainty normalization, lithology classification, and instrument cleaning. Please see section 3 in the revised manuscript. In addition, we supplemented our database with isotopic ratios and uncertainties (including $^{206}Pb/^{238}U$, $^{207}Pb/^{235}U$, $^{207}Pb/^{206}Pb$, and $^{208}Pb/^{232}Th$), as well as zircon reference materials (such as 91500 and GJ-1). Please see the database version v2 in the Zenodo repository (https://doi.org/10.5281/zenodo.8040079).

RC: Inconsistency of data:

1. The "Method" field mixes analytical methods with instruments; sometimes only a reference is cited. These should be separated and you should use a controlled list for both the analytical methods and the instruments: for example, there are >10 different spellings for ICP-MS. What is the difference between null values and those labelled "unmentioned"?

2. Fig 1, and the text in general, gives the impression that you have location information for all records. However, coordinates are missing for many entries in the data sheets.

3. The reference file should also include DOI, title, name of co-authors to guarantee unique identification of the data source. These citations should be included in the reference list to this manuscript.

AR: Thank you for your comments. Please find below a point-by-point response to the questions you listed:

1. Sorry for the confusion. This confusion arises due to that the initials for both the analytical method ("Mass Spectrometry") and instrument ("Mass Spectrometer") are "MS". We will replace the "Method" field with "Instrument". Second, there are different spellings for ICP-MS because the original literature wrote that way. We would like to keep these different spellings ("different" instruments) to provide researchers with more options. Third, we will further clean the "Instrument" field, such as addressing the "unmentioned" label and cited references.

2. We will give a clear statement on the null values of location information. Null values are inevitable because the original literature didn't provide associated information. However, the rest information is still helpful for some studies.

3. We will add DOI and title in the reference file. We need to declare that some dissertations and old papers (especially papers in Chinese journals) don't have a DOI. But, the unique identification of the data source can still be guaranteed by other information we provided. We prefer to put the source references in supplementary materials or the Zenodo repository instead of the manuscript. Because the original data sources have more than 10,000 papers, which will take more than 400 pages if put in the manuscript reference list.

AC: Changes:

1. To avoid confusion, we used "instrument" instead of "method" to describe dating approaches in the entire revised manuscript. In our updated database (version v2) in the Zenodo repository (https://doi.org/10.5281/zenodo.8040079), we replaced the field "Method" with "Instrument" and cleaned the corresponding contents. We are sorry for what we said in the first point of this "Author Response" (AR). We agree that the different spellings for ICP-MS do bring difficulties for future analysis. There, we cleaned this "Instrument" field by grouping all the instruments into four major groups: LA-ICP-MS, SHRIMP, SIMS, and TIMS. We will keep the old database (version v1) in the Zenodo repository for readers who are interested in in-depth research on dating instruments.

2. We updated Table 1 to describe the proportion of non-null values. We also added some related sentences (lines 186-188 in the revised manuscript with track-changes).

3. We updated the "References" file by adding fields on the literature title, first page, last page, DOI, and URL. Please see Table 2 and lines 278-283 in the revised manuscript with track-changes. Also, please see the "References" file of our updated database (version v2) in the Zenodo repository (https://doi.org/10.5281/zenodo.8040079).

RC: Sustainability of the database: is this a curated database that will be maintained and updated? If so, over what timeframe will it be maintained? If not, have there been any attempts to integrate your work with existing, curated compilations such as those of EarthChem (https://earthchem.org/), GEOROC (https://georoc.eu/), Martin et al. (2022, https://doi.org/10.1038/s41597-022-01730-7 and https://doi.org/10.25625/FWQ7DT)?

AR: This is a good point. At present, our Zircon database is not a curated database, and we do not have a specific maintenance and update plan for it. While the maintenance and update work is a possibility that may be considered in future projects, we believe that the current format of the database does not hinder its research potential.

We have not considered the integration of our database with other compilations yet because we prefer that our database maintains its independent existence. The EarthChem and GEOROC are undoubtedly great curated compilations. However, the Zenodo repository is also a good platform for promoting open science and sharing our database with the research community.

AC: We have updated our zircon database in the Zenodo repository to address your

concerns (database version v2: https://doi.org/10.5281/zenodo.8040079). As you can see, the Zenodo repository is also a good platform for open science. It is convenient to store, update, manage, and share data. You can cite all versions by using the DOI 10.5281/zenodo.7387566. You can also cite a specific version using the corresponding DOI (version v1: 10.5281/zenodo.7387567; version v2 10.5281/zenodo.8040079).

RC: Incomplete referencing:
1. Of other zircon geochronology compilations (e.g. EarthChem, GEOROC, Martin et al., 2022). How much overlap exists to these previous compilations? Equally, how many data are missing?
2. Of scientific literature, including statistical treatment of oversampling/sampling bias, which should be applied to your database before any geological interpretations are drawn (e.g. Keller & Schoene, 2012: https://doi.org/10.1038/nature11024; Mehra et al., 2021: https://doi.org/10.1130/GSATG484A.1)

AR: We appreciate your comments. Please find below a point-by-point response to the questions you listed:
1. Our zircon database is independent of other compilations. We began the data construction in 2017 and we didn't refer to other compilations when constructing our own. Since the forms of the databases are different, it is difficult to compare them one by one to check the overlap or missing data. There might be some overlap since we might collect the same literature. Nonetheless, our database does have unique advantages. For example, we collected a large amount of data in Chinese literature, which is difficult for non-Chinese scholars to obtain. We believe the diversity of databases can provide more options for future research.
2. We will address the sampling bias in the revised manuscript and use the new results for geological interpretations. In the meantime, we want to keep the results using raw data in the main text or supplementary materials for reference because they display the original characteristics of the zircon data. It is also possible that researchers can use our raw data to explore more advanced ways to deal with the biased sampling issue in the future.

AC: We addressed the sampling bias issue using the methods of inverse proximity weighting, bootstrapping resampling, and Monte Carlo simulation referring to the papers you suggested (Keller & Schoene, 2012; Mehra et al., 2021). Please see Figures 4 and 5 and lines 397-399 in the revised manuscript with track-changes, and Figure S5 and Methods in the revised supplementary materials.

RC: The discussion & scientific interpretation are very superficial, with language that is both too informal and very pompous. Previous work on this topic is not discussed in sufficient detail. As this is a submission to ESSD, I believe that much of Sections 3 and 4 goes beyond the scope of a data journal and could be removed. My recommendation would be to instead focus primarily on Section 4.5 and ensure that discussion of previous literature in this section is comprehensive, detailed and

accurate.

AR: Sorry for our sketchy discussion and poor language. We didn't write the discussion and scientific interpretation in detail to avoid extensive interpretations of data. Perhaps there was a little deviation in our understanding of the scope of ESSD. As you suggested, we will focus on Section 4.5 in the revised manuscript, adding comprehensive, detailed, and accurate discussion. We will remove Sections 4.1 to 4.4 and summarize the content in one or two paragraphs instead. Additionally, we will abridge Section 3 as much as possible to ensure that data description is the main purpose of the paper. However, we want to keep an abridged Section 3 in the manuscript because this section is necessary, which intuitively presents the basic characteristics of the zircon data. Finally, we will find an advanced editing service to improve the language.

AC: According to your suggestions, we did abridgment and supplement. We added a new section 3 on data cleaning, so the original sections 3 and 4 correspond to the revised sections 4 and 5. The abridged section 4 only contains two sub-sections, namely sub-section 4.1 on dating uncertainty and sub-section 4.2 on temporal and spatial characteristics, with paragraphs significantly shortened. The out-of-scope discussion contents (the original sub-sections 4.1-4.4) were shortened to one concise sub-section with two paragraphs (the revised sub-section 5.1). In addition, we expanded the discussion on the biased sampling issues (the revised sub-section 5.2). We hope these revisions address your concerns.

---

## Referee Report (RR2)

The revised manuscript by Wu et al. is much improved compared to the original submission and the authors have taken care to respond to all comments raised by both reviewers. I recommend the publication of this revised manuscript pending a few minor corrections as detailed below.

Kind regards,
Marthe Klöcking

**Detailed comments:**
1. The article is due to be published with a CC BY 4.0 Attribution license; it is unclear what license will be linked to the Zenodo dataset. Has it been checked that all licenses of the data sources in the compilation have been honoured and that CC BY 4.0 is appropriate?
2. I accept that you do not wish to compare the content of your data compilation with existing global compilations at such as those of EarthChem (https://earthchem.org/), GEOROC (https://georoc.eu/; e.g. https://doi.org/10.25625/SGFTFN/AKMJG2), Martin et al. (2022, https://doi.org/10.1038/s41597-022-01730-7 and https://doi.org/10.25625/FWQ7DT). However, I do expect these previous efforts to be mentioned and referenced in your manuscript. Please include these references, e.g. in Sections 1 or 2 (Introduction/Database).
3. As you rightly point out, a huge benefit of this dataset lies in the large number of Chinese data that have been compiled. This fact is stated at the beginning of the manuscript but also deserves to be emphasised again in the later discussion. Conversely, global coverage for the rest of the world is comparatively sparse. While you discuss sampling bias and present methods for statistical resampling, I think it would be helpful to honestly present this regional disparity throughout the manuscript. In numbers, what proportion of the 2 million geochronological records lie outside of China? A quick check of zircon data available through the EarthChem Portal shows that the sample distribution in other, existing global compilations (~500,000 records across GEOROC, EarthChem, GANSEKI, NAVDAT and MetPetDB; see screenshot below) could perhaps be a valuable resource to complement the data presented here, prior to any statistical analysis.

[Figure]

4. Typos & grammatical errors:

L10: word missing after "geochronological", e.g. "geochronological records"

L12: Please remove unnecessary and subjective statement "and is by far the largest geochronological database to our knowledge"

L12: "complied" should be "compiled"

L23: reference "(Becker, 2007)" seems out of place

L38: "all of" rather than "all of the"

L56: add reference to Martin et al.; EarthChem and GEOROC compilations

L58: please rephrase "dating data points"

L59: I would use "techniques" rather than "instruments" here

L61: remove "the" before "temporally and spatially"

L150-152: the use of "etc" is jarring. Rewrite as "includes, for example, [A, B, C, …]"

Figure 6: what does the colour scale represent? Please add label

L277: rephrase "largest known database" into a more quantitative and less subjective statement

---

## Author Response (AR2)

**Point-by-point response for major revision**

**A global zircon U–Th–Pb geochronological database**

Yujing Wu et al.

*Earth Syst. Sci. Data Discuss.*, https://doi.org/10.5194/essd-2023-20

RC: Referee Comment, AR: Author Response

Dear Editor,

Thank you very much for your precious time and effort spent on our manuscript and dataset. The referees' comments and suggestions were greatly helpful.

We revised the manuscript as the referees suggested. The supplement seemed good and was not changed this time. About the Zircon database, we added an *Access* column which contains access type or license information in the "References" file in the Zenodo repository. I am very grateful for your professional help on the license issue. Currently, the Zircon database is under "restricted access". Once the paper is accepted and/or ready to be published, we will change it to "open access" under the license of Creative Commons Attribution 4.0 International (CC BY 4.0).

Please find below a point-by-point reply to all referee comments and the corresponding revisions.

Many thanks and kind regards,
Yujing Wu (on behalf of the author team)

**Author Response to Referee #1**

**A global zircon U–Th–Pb geochronological database**

Yujing Wu et al.

*Earth Syst. Sci. Data Discuss.*, https://doi.org/10.5194/essd-2023-20

RC: Referee Comment, AR: Author Response

Dear referee,

Thank you very much for your positive response, and for your precious time and effort spent reviewing the manuscript and the dataset. Your suggestions are of great help.

Please find a point-by-point reply below.

Kind regards,

Yujing Wu (on behalf of the author team)

**Comments and responses:**

**RC: General Comments:**
> The revised manuscript is much improved and nearly ready for publication. Appropriately, it now discusses areas of research that require further investigation. Of course, this new database will be helpful in resolving these uncertainties. I was critical of earlier versions of the database (including understandings gained from discussions with the authors prior to this submission), but now believe the database is structured in such a way that the research community can use it optimally. I will be among those who use and study this database. There are still a few minor revisions, mostly grammatical, that will improve the manuscript, which are listed below. After making these minor revisions, I recommend publication. Overall, this is excellent work.

**AR: General Responses:**
> Thank you very much for your support. We did corresponding revisions as you suggested. Please find a point-by-point reply below.

**RC: Specific Comments:**

RC: **Line 61:** Poor grammar and a confusing statement. Suggest revising the phrase "This database may provide a more comprehensive and objective chronology data source both the temporally and spatially" to state "This database provides a comprehensive source of geochronological data, both temporally and spatially, ..."

AR: Implemented. Please see lines 70-71 in the revised manuscript with track-changes.

RC: **Lines 155-156:** I have questions about the method Keller and Schoene (2012) use for minimizing sampling bias, and suspect that it might sometimes give inappropriate results. There is no need for the authors to make any revisions

because my investigations into this matter remain incomplete. In fact, this new database might help resolve this problem.

AR: Thank you for your open perspective on this point and your support of this zircon database. We are looking forward to new approaches to minimizing sampling bias. Glad to hear that our database may help.

RC: **Line 196:** The phrase "absolute zircon densities" might be misleading and confused with "absolute sampling densities". To eliminate this likely confusion, the authors should rephrase this to state "absolute age counts per 50 or 100-Myr bin"... but only if this is the author's intent.

AR: Sorry for the confusion. The following explanation was added after this confusing sentence.

The "zircon density" here refers to age counts per grid which is bounded by longitudes of 4° length and latitudes of 2° width during the peak periods (50 or 100 Myr).

Please see lines 223-224 in the revised manuscript with track-changes.

RC: **Lines 223-224:** Suggest re-phrasing "much zircon production" to state "amplified zircon production".

AR: Implemented. Please see line 258 in the revised manuscript with track-changes.

RC: **Line 224:** Suggest replacing "zircon production series" to state "zircon production time series".

AR: Implemented. Please see line 259 in the revised manuscript with track-changes.

RC: **Lines 250-254:** Suggest removing the sentence: "Although some important magmatic and metamorphic activities for crustal evolution (such as orogenic belts and subduction zones) were concentrated in certain regions and periods, the distributions of zircon ages were remarkably similar regardless of depth or height (Parman, 2015)." Parman's method of analyzing U-Pb age distributions, as a function of depositional-age intervals, is completely unrelated to the grid-area method of Puetz et al. (2017). For this reason, it is confusing why this sentence was even placed here.

AR: Sorry for the improper citation. We deleted this citation and the first half of the sentence. We kept "the distributions of zircon ages were remarkably similar regardless of depth or height" to explain to a broader audience why the grid-area method works well. Please see line 289 in the revised manuscript with track-changes.

RC: **Line 271-272:** No revisions necessary. I agree with this statement: "New approaches can be explored to address biased sampling issues, and this zircon database provides great experimental materials." ... Very good, I agree these questions are unresolved and require further investigation.

AR: Thanks a lot for your support.

**Author Response to Referee #2**

**A global zircon U–Th–Pb geochronological database**

Yujing Wu et al.

*Earth Syst. Sci. Data Discuss.*, https://doi.org/10.5194/essd-2023-20

RC: Referee Comment, AR: Author Response

Dear referee,

Thank you sincerely for your positive response, and for dedicating your valuable time and effort to reviewing both the manuscript and the dataset. Your suggestions and concerns are of great help.

Please find our point-by-point reply below.

Kind regards,
Yujing Wu (on behalf of the author team)

**Comments and responses:**

**RC: General Comments:**

> The revised manuscript by Wu et al. is much improved compared to the original submission and the authors have taken care to respond to all comments raised by both reviewers. I recommend the publication of this revised manuscript pending a few minor corrections as detailed below.

**AR: General Responses:**

> We greatly appreciate your supportive comments and feedback. We have revised the manuscript and updated the zircon database as you suggested. Please see the point-by-point response below.

**RC: Detailed Comments:**

RC: The article is due to be published with a CC BY 4.0 Attribution license; it is unclear what license will be linked to the Zenodo dataset. Has it been checked that all licenses of the data sources in the compilation have been honoured and that CC BY 4.0 is appropriate?

AR: Great question! Currently, our Zenodo dataset is a "restricted access" dataset. Once the paper is accepted and/or ready to be published, we will change it to "open access" under the license of Creative Commons Attribution 4.0 International (CC BY 4.0).

Thanks a lot for your comments. We checked all access types or licenses of the data sources. 11,284 out of 11,571 references are "free-to-download" and the other 291 references are "open access" (licenses including CC BY, CC BY-NC, CC BY-NC-ND, CC BY-NC-SA, and other open-access types for specific journals). We

added an *Access* column which contains access type or license information in the "References" file. Please check the updated Zenodo repository (version 2.1) for more details.

The Editor Elger helped me ask a professional colleague who said that extracting data from articles is not violating any copyright and I am not bound to the license of the article. There is no issue with publishing our database with an open license. Although we didn't cite the original sources one by one, we used the "References.xlsx" to give credit to the original work.

Thank you so much for this question. I learned a lot about open access through it.

RC: I accept that you do not wish to compare the content of your data compilation with existing global compilations at such as those of EarthChem (https://earthchem.org/), GEOROC (https://georoc.eu/; e.g. https://doi.org/10.25625/SGFTFN/AKMJG2), Martin et al. (2022, https://doi.org/10.1038/s41597-022-01730-7 and https://doi.org/10.25625/FWQ7DT). However, I do expect these previous efforts to be mentioned and referenced in your manuscript. Please include these references, e.g. in Sections 1 or 2 (Introduction/Database).

AR: Sorry for our carelessness. We added some sentences about these references in the Introduction. Please see lines 62-66 in the revised manuscript with track-changes.

RC: As you rightly point out, a huge benefit of this dataset lies in the large number of Chinese data that have been compiled. This fact is stated at the beginning of the manuscript but also deserves to be emphasised again in the later discussion. Conversely, global coverage for the rest of the world is comparatively sparse. While you discuss sampling bias and present methods for statistical resampling, I think it would be helpful to honestly present this regional disparity throughout the manuscript. In numbers, what proportion of the 2 million geochronological records lie outside of China? A quick check of zircon data available through the EarthChem Portal shows that the sample distribution in other, existing global compilations (~500,000 records across GEOROC, EarthChem, GANSEKI, NAVDAT and MetPetDB; see screenshot below) could perhaps be a valuable resource to complement the data presented here, prior to any statistical analysis.

[Figure]

AR: Great points.

1) We mentioned the large numbers of Chinese and European data in the Discussion (Section 5.2). Please see line 282 in the revised manuscript with track-changes.

2) We added a quantitative explanation of the regional disparity in the sections of Database and Discussion. Since our Zenodo database is not a curated database and doesn't provide a GUI (Graphical User Interface) tool to show regional disparity quantitatively, we did simple statistics using the GPS data. For zircon records with GPS information (~1.6 million records), data sampled from China and outside China account for ~42% and 58%, respectively. Please see lines 122-124 and 282-284 in the revised manuscript with track-changes.

RC: Typos & grammatical errors:
RC: L10: word missing after "geochronological", e.g. "geochronological records"

AR: We changed "geochronological" to "geochronology". Please see line 10 in the revised manuscript with track-changes.

RC: L12: Please remove unnecessary and subjective statement "and is by far the largest geochronological database to our knowledge"

AR: Removed. Please see line 12 in the revised manuscript with track-changes.

RC: L12: "complied" should be "compiled"

AR: Implemented. Please see line 12 in the revised manuscript with track-changes.

RC: L23: reference "(Becker, 2007)" seems out of place

AR: Moved to the end of the sentence. Please see line 25 in the revised manuscript with track-changes.

RC: L38: "all of" rather than "all of the"

AR: Implemented. Please see line 38 in the revised manuscript with track-changes.

RC: L56: add reference to Martin et al.; EarthChem and GEOROC compilations

AR: We added the citations and some sentences on these studies. Please see lines 62-66 in the revised manuscript with track-changes.

RC: L58: please rephrase "dating data points"

AR: Rephrased as "dating records". Please see line 68 in the revised manuscript with track-changes.

RC: L59: I would use "techniques" rather than "instruments" here

AR: Changed to "techniques". Please see line 69 in the revised manuscript with track-changes.

RC: L61: remove "the" before "temporally and spatially"

AR: Removed. Please see line 71 in the revised manuscript with track-changes.

RC: L150-152: the use of "etc" is jarring. Rewrite as "includes, for example, [A, B, C, …]" Figure 6: what does the colour scale represent? Please add label

AR: 1) We revised the text using "for example". Please see lines 170-172 in the revised manuscript with track-changes.
2) We updated the caption of Figure 6 to explain the color scale. Please see lines 248-250 in the revised manuscript with track-changes.

RC: L277: rephrase "largest known database" into a more quantitative and less subjective statement

AR: We deleted "the largest known" and added an "a". Since the next sentence describes the amount of the records (~ 2 million), we didn't add other quantitative phrases. Please see lines 322-323 in the revised manuscript with track-changes.

---

## Author Response (AR3)

**Point-by-point response for minor revision**

A global zircon U–Th–Pb geochronological database

Yujing Wu et al.

*Earth Syst. Sci. Data Discuss.*, https://doi.org/10.5194/essd-2023-20

RC: Referee Comment, AR: Author Response

Dear Editor,

Thank you very much for your precious time and effort spent on our manuscript and dataset. We followed your kind advice and did the corresponding revision.

Since the latest referee comment (report 21 Sep 2023) suggested that the manuscript should be "accepted as is", the only revision we did this time was that we removed the access barrier of the Zircon database in the Zenodo repository and added a license of CC BY 4.0. The DOI of the database is unchanged, so we didn't change the manuscript and supplement this time.

Please find below a reply to the referee's comments and the corresponding revisions.

Many thanks and kind regards,
Yujing Wu (on behalf of the author team)

**Author Response to Referee #2**

A global zircon U–Th–Pb geochronological database

Yujing Wu et al.

*Earth Syst. Sci. Data Discuss.*, https://doi.org/10.5194/essd-2023-20

RC: Referee Comment, AR: Author Response

Dear referee,

Thank you sincerely for your positive response, and for dedicating your valuable time and effort to reviewing both the manuscript and the dataset.

Kind regards,
Yujing Wu (on behalf of the author team)

**Comments and responses:**

**RC: Comments:**

    For final publication, the manuscript should be "accepted as is".

**AR: Responses:**

    We greatly appreciate your supportive comments.